# Fitness effects of altering gene expression noise in *Saccharomyces cerevisiae*

**Fabien Duveau[1,2], Andrea Hodgins-Davis[1], Brian PH Metzger[1,3], Bing Yang[4], Stephen Tryban[1], Elizabeth A Walker[1], Tricia Lybrook[1], Patricia J Wittkopp[1,4]\***

[1]Department of Ecology and Evolutionary Biology, University of Michigan, Ann Arbor, United States; [2]Laboratoire Matière et Systèmes Complexes, CNRS UMR 7057, Université Paris Diderot, Paris, France; [3]Department of Ecology and Evolution, University of Chicago, Chicago, United States; [4]Department of Molecular, Cellular and Developmental Biology, University of Michigan, Ann Arbor, United States

**Abstract** Gene expression noise is an evolvable property of biological systems that describes differences in expression among genetically identical cells in the same environment. Prior work has shown that expression noise is heritable and can be shaped by selection, but the impact of variation in expression noise on organismal fitness has proven difficult to measure. Here, we quantify the fitness effects of altering expression noise for the *TDH3* gene in *Saccharomyces cerevisiae*. We show that increases in expression noise can be deleterious or beneficial depending on the difference between the average expression level of a genotype and the expression level maximizing fitness. We also show that a simple model relating single-cell expression levels to population growth produces patterns consistent with our empirical data. We use this model to explore a broad range of average expression levels and expression noise, providing additional insight into the fitness effects of variation in expression noise.
DOI: https://doi.org/10.7554/eLife.37272.001

**\*For correspondence:**
wittkopp@umich.edu

## Introduction

Gene expression is a dynamic process that results from a succession of stochastic biochemical events, including availability of transcription factors, binding of transcription factors to promoter sequences, recruitment of transcriptional machinery, transcriptional elongation, mRNA degradation, protein synthesis, and proteolysis. These events cause the expression level of a gene product to differ even among genetically identical cells grown in the same environment (*Elowitz et al., 2002*; *Chong et al., 2015*). This variability in gene expression is known as 'expression noise' and is under genetic control (*Raser et al., 2004*; *Sanchez and Golding, 2013*), with heritable variation causing differences in noise among genes (*Newman et al., 2006*) and genotypes (*Murphy et al., 2007*; *Hornung et al., 2012*; *Fehrmann et al., 2013*; *Sharon et al., 2014*; *Liu et al., 2015*).

Because gene expression noise is heritable and variable within populations, it can evolve in response to natural selection if it affects fitness. Indeed, prior studies have suggested that expression noise can be either beneficial or deleterious depending on the context (reviewed in *Viney and Reece, 2013*; *Richard and Yvert, 2014*; *Liu et al., 2016*). For example, *Metzger et al. (2015)* provides evidence that increased expression noise can be selected against in natural populations, presumably because elevated noise increases the probability that a given cell produces a suboptimal level of protein expression (*Wang and Zhang, 2011*; *Duveau et al., 2017a*). Consistent with this hypothesis, a negative correlation exists at the genomic scale between the expression noise of genes

**eLife digest** Single-celled organisms that reproduce by dividing, like yeast, can create whole populations of genetically identical cells. However, some differences will exist among such cells, even when they have all experienced the same environment. These differences are known as "noise". By definition, noise is not caused by differences in DNA sequence, but some DNA sequences are noisier than others (i.e. they cause more differences among cells). Because the amount of noise can be under genetic control, noise could evolve due to natural selection.

Scientists often study noise at the level of gene expression – in other words, how many RNA or protein molecules are produced from each gene within each cell. Prior work has suggested that this type of noise can affect how often individual cells divide in a population, which is a component of that population's fitness. Yet directly measuring these effects has proven challenging. Different studies have in the past reached opposite conclusions about whether a change in gene expression noise would increase or decrease fitness.

One major reason for the lack of clear results is that most mutations that alter gene expression noise also alter the average level of expression of that gene. To find DNA sequences that produced the same average amount of protein but different levels of expression noise, Duveau et al. compared the effects of hundreds of mutations in the DNA sequence regulating the expression of a gene in baker's yeast. Experiments focused on 43 DNA sequences then showed that increased expression noise could either speed up or slow down the growth of the population by affecting how long it took each cell to divide. More specifically, the effects of increasing expression noise depended on the average amount of protein produced among the cells in the population. If the average expression level was close to the optimum amount at which cells divided as fast as possible, increasing expression noise reduced the growth of the whole population. If, however, the average protein level caused cells to divide slower than their maximum rate, increasing expression noise resulted in faster growth of the population as a whole.

Duveau et al. explain their results as follows: more expression noise in a population that is already making the optimal amount of protein can reduce fitness because it increases the fraction of that population making a suboptimal amount of the protein. However, when the average expression level is not optimal, more expression noise would mean more cells producing an amount of protein that is closer to the optimum and thus having higher fitness.

These findings provide conceptual tools needed to understand how genetic variation affecting expression noise evolves. They could also help understand how expression noise might contribute to biological processes that depend upon cell division, such as diseases like cancer.

DOI: https://doi.org/10.7554/eLife.37272.002

and their dosage sensitivity (*Fraser et al., 2004*; *Batada and Hurst, 2007*; *Lehner, 2008*; *Keren et al., 2016*). However, because the optimal level of gene expression can vary among environments, high gene expression noise has been suggested to be beneficial if it can produce individuals with phenotypes that are better adapted to a new environment than individuals produced with low gene expression noise. For instance, noise in gene expression can allow a small fraction of cells to survive when confronted with stressful environmental conditions (*Blake et al., 2006*; *Fraser and Kaern, 2009*; *Ito et al., 2009*; *Levy et al., 2012*; *Viney and Reece, 2013*; *Liu et al., 2015*; *Wolf et al., 2015*). Consistent with this idea, a genomic screen in yeast found that plasma-membrane transporters involved in cell-environment interactions displayed elevated expression noise in yeast (*Zhang et al., 2009*). Theoretical work also suggests the existence of cost-benefit tradeoffs that can make expression noise either beneficial or deleterious under different circumstances (*Tănase-Nicola and ten Wolde, 2008*).

Despite a growing body of evidence that selection has acted on expression noise for many genes, direct measurements of how changes in expression noise impact fitness remain scarce (*Liu et al., 2016*). A major reason for this scarcity is that most mutations that alter gene expression noise also alter average expression level (*Newman et al., 2006*; *Hornung et al., 2012*; *Carey et al., 2013*; *Sharon et al., 2014*), making it difficult to disentangle the fitness effects of changing expression noise and average expression level. Here, we directly estimate the effects of changing expression

noise on fitness independently from changes in average expression level for the *TDH3* gene of *Saccharomyces cerevisiae*.

*TDH3* encodes an isozyme of the yeast glyceraldehyde-3-phosphate dehydrogenase (GAPDH) involved in glycolysis and gluconeogenesis (*McAlister and Holland, 1985*) as well as transcriptional silencing (*Ringel et al., 2013*), RNA-binding (*Shen et al., 2014*) and possibly antimicrobial defense (*Branco et al., 2014*). Variation in this gene's promoter affecting expression noise has previously been shown to be a target of selection in natural populations (*Metzger et al., 2015*). To assess the impact of changes in expression noise on fitness at different expression levels, we generated mutant alleles of the *TDH3* promoter that covered a broad range of average expression levels and expression noise. We find that increases in expression noise are detrimental when the average expression level of a genotype is close to the fitness optimum, but beneficial when the average expression level of a genotype is further from this optimum. This pattern was reproduced by a simple computational model that links expression in single cells to their doubling time to predict population fitness. We used this individual-based model to explore the fitness effects of a broader combination of average expression levels and expression noise than were explored empirically, showing that not only do the fitness effects of changing expression noise depend on the average expression level, but that the fitness effects of changing average expression level also depend upon the amount of expression noise.

## Results and discussion

### Generating variation in expression noise independent of average expression level

To disentangle the effects of changes in average expression level and expression noise on fitness, we examined a set of *TDH3* promoter ($P_{TDH3}$) alleles with a broad range of activities. For each allele, we measured the average expression level and expression noise by cloning the allele upstream of a yellow fluorescent protein (YFP) coding sequence, integrating this reporter gene ($P_{TDH3}$-*YFP*) into the *HO* locus, and quantifying fluorescence in living cells using flow cytometry in six replicate populations per genotype (*Figure 1A*). The fluorescence value of each cell was transformed into an estimated mRNA level (*Figure 1A*) based on the relationship between fluorescence and YFP mRNA abundance (*Figure 1B,C*). The average expression level of a genotype was then calculated by averaging the median values from the six replicates (*Figure 1A*) and expressing this value as a percent change from the wild type allele. Expression noise was calculated for each replicate as the variance divided by the median expression among cells, a measure of noise strength similar to the Fano factor (*Sanchez and Golding, 2013*). The expression noise of each genotype was then calculated by averaging the noise strength from the six replicate populations, and this value was expressed as a percent change from the wild type allele. The main conclusions of this study are robust to the choice of noise metric, as shown in supplementary figures using three alternative metrics of noise.

Effects of 236 point mutations in the *TDH3* promoter, including mutations in RAP1 and GCR1 transcription factor binding sites (TFBS), have previously been described that cause a wide range of average expression levels and expression noise values (*Metzger et al., 2015*). But average expression level and expression noise strongly co-vary among these alleles (*Metzger et al., 2015*), making them insufficient for separating the effects of changes in average expression level and expression noise on fitness. We therefore sought to construct additional promoter alleles that showed a different relationship between average expression level and expression noise. First, we inserted a recognition motif for the GCN4 transcription factor at ten different positions in the *TDH3* promoter because this TFBS was previously found to affect the relationship between expression level and expression noise (*Sharon et al., 2014*). However, the insertion of GCN4 binding sites into $P_{TDH3}$ did not show the expected departure from the relationship between expression level and expression noise observed for mutations in GCR1 and RAP1 TFBS (*Figure 1—figure supplement 1*). We next mutated the $P_{TDH3}$ TATA box because previous studies showed that TATA box mutations confer lower expression noise for a given expression level when compared to other types of promoter alterations (*Blake et al., 2006*; *Mogno et al., 2010*; *Hornung et al., 2012*). We generated 112 alleles of the *TDH3* promoter that had between one and five random mutations in the TATA box sequence, which caused the expected lower levels of expression noise than TFBS mutant alleles with similar

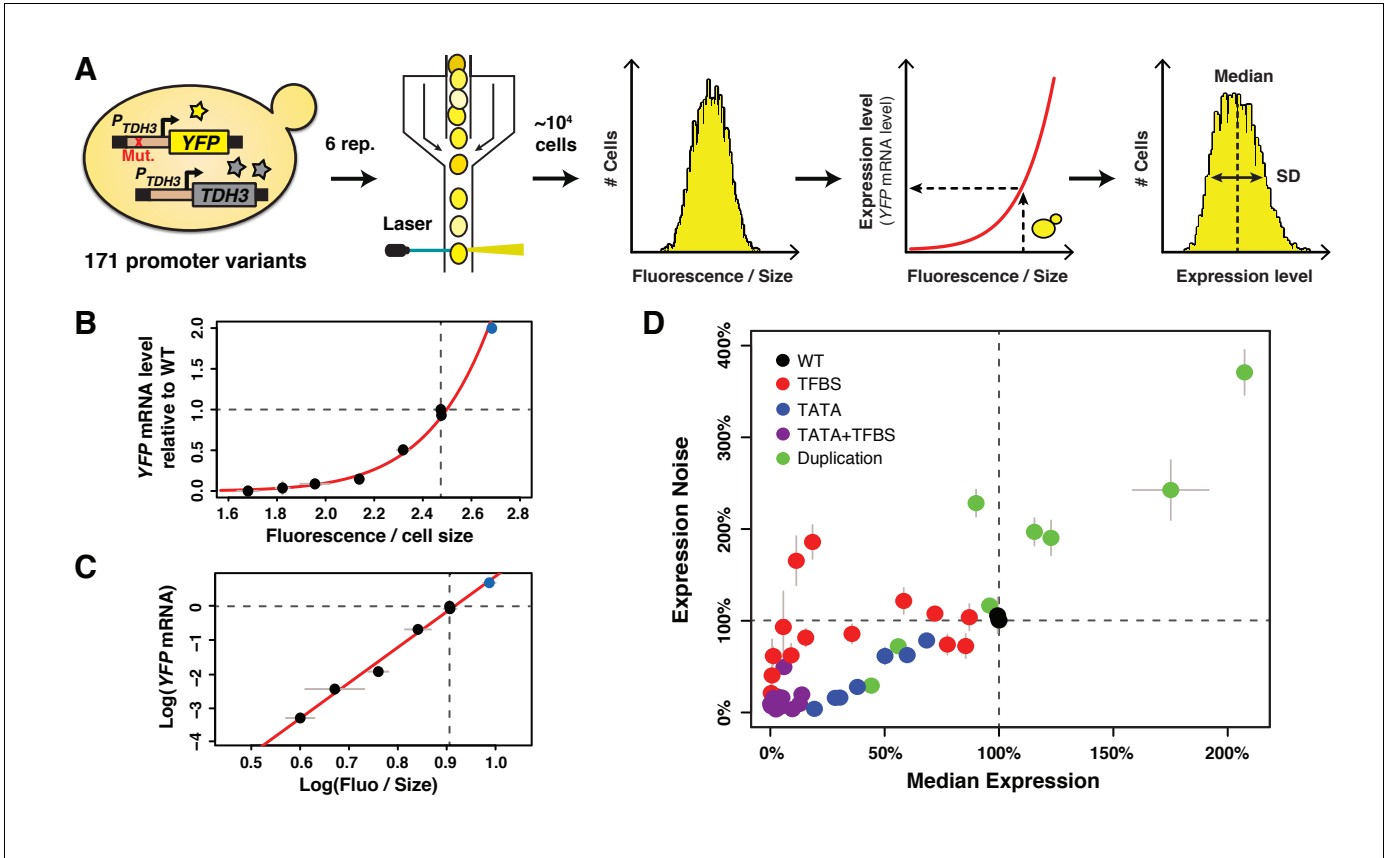

**Figure 1.** A collection of *TDH3* promoter alleles with incompletely correlated effects on average expression level and expression noise. (**A**) Overview of experimental design used to quantify expression. The transcriptional activity of 171 different variants of the *TDH3* promoter ($P_{TDH3}$) inserted upstream of the *YFP* coding sequence was quantified using flow cytometry. After growth of six independent samples in rich medium (YPD) for each promoter variant, fluorescence intensity relative to cell size (forward scatter) was measured for ~10,000 individual cells and transformed into *YFP* mRNA estimates using the function shown in (**B**), allowing characterization of both the median and the standard deviation of expression of the reporter gene. (**B**) Non-linear relationship between *YFP* mRNA level and fluorescence intensity divided by cell size measured on a BD Accuri C6 flow cytometer. (**C**) Linear relationship between the logarithm of *YFP* mRNA level and the logarithm of fluorescence intensity divided by cell size. (**B–C**) *YFP* mRNA level was quantified by pyrosequencing and fluorescence intensity by flow cytometry in three biological replicates of eight strains expressing *YFP* under different variants of $P_{TDH3}$. Fluorescence intensity was normalized by cell size as described in the Materials and methods section. The red line is the best fit of a function of shape $\log(y) = a \times \log(x) + b$ to the data, with $a = 10.469$ and $b = -9.586$. The blue dot represents a strain with two copies of the wild type $P_{TDH3}$-*YFP* reporter. Data are available in *Figure 1 – source data 1*. (**D**) Median expression level and expression noise (noise strength: variance divided by median expression) for 43 $P_{TDH3}$ alleles. These alleles were chosen to cover a broad range of median expression level and expression noise with an incomplete correlation between these two parameters. Colors represent different types of promoter mutations. Data are available in *Source data 1*. (**B–D**) Dotted lines show the activity of the wild type *TDH3* promoter. Error bars are 95% confidence intervals calculated from at least four replicates for each genotype and are only visible when larger than dots representing data.

DOI: https://doi.org/10.7554/eLife.37272.003

The following source data and figure supplements are available for figure 1:

**Source data 1.** Parallel measurements of fluorescence levels by flow cytometry and of *YFP* mRNA levels by pyrosequencing.
DOI: https://doi.org/10.7554/eLife.37272.006
**Figure supplement 1.** Median expression level and expression noise conferred by 171 variants of the *TDH3* promoter using four different metrics of noise.
DOI: https://doi.org/10.7554/eLife.37272.004
**Figure supplement 1—source data 1.** Expression data for an initial set of 171 *TDH3* promoter alleles.
DOI: https://doi.org/10.7554/eLife.37272.005

average expression levels (*Figure 1D*). We then combined mutations in the TATA box, GCR1 TFBS and/or RAP1 TFBS to further increase the range of expression phenotypes. Finally, we constructed alleles containing two tandem copies of the $P_{TDH3}$-YFP reporter gene with or without mutations in the $P_{TDH3}$ sequence to sample expression levels closer to and above the wild-type allele. These mutant alleles captured a much greater range of mean expression and expression noise than *TDH3* promoter alleles segregating in natural populations (*Metzger et al., 2015*; *Duveau et al., 2017a*) and allowed us to more fully explore the relationship between mean, noise and fitness than would be possible using naturally occurring variation alone.

From this collection of 171 *TDH3* promoter alleles (*Figure 1—figure supplement 1*, *Figure 1—figure supplement 1—source data 1*), we selected 43 alleles (*Source data 1*) to study the fitness effects of changes in average expression level and expression noise of the native *TDH3* gene. The average expression level conferred by these 43 $P_{TDH3}$ alleles (including the wild type allele of $P_{TDH3}$) ranged from 0% to 207% of the wild type allele and the expression noise ranged from 3% to 371% of the wild type allele (*Figure 1D*). Most importantly, this set of alleles showed variation in expression noise independent of expression level at expression levels between 0% and 125% of the wild type allele (*Figure 1D*).

## Fitness effects of changing average *TDH3* expression level

To measure the fitness effects of changing *TDH3* expression, we introduced each of these 43 $P_{TDH3}$ alleles upstream of the *TDH3* coding sequence at the native *TDH3* locus and performed competitive growth assays similar to those described in *Duveau et al. (2017a)* (*Figure 2A*). For each of the eight $P_{TDH3}$ alleles that contained a duplication of the $P_{TDH3}$-YFP reporter gene, we created a duplication of the entire *TDH3* gene with the two corresponding $P_{TDH3}$ alleles. We also included a strain with a deletion of the promoter and coding sequence of *TDH3* to sample a *TDH3* expression level of 0%. Prior studies have found that deletion of *TDH3* causes a moderate decrease in fitness in glucose-based media: −5% in *Pierce et al. (2007)* and −6.8% in *Duveau et al. (2017a)*. Each strain tested was marked with YFP and grown competitively for 30 hr (~21 generations) with a reference genotype marked with a green fluorescent protein (GFP) (*Figure 2A*). Competitive fitness was determined from the rate of change in genotype frequencies over time and averaged across at least six replicate populations for each genotype tested (*Figure 2B*). The relative fitness of each strain was then calculated by dividing the competitive fitness of that strain by the competitive fitness of the strain with the wild type allele of *TDH3* (*Source data 1*). This protocol provided a measure of fitness with an average 95% confidence interval of 0.2%. We then related these measures of relative fitness to the expression of the reporter gene driven by these $P_{TDH3}$ alleles at the *HO* locus. Expression of this reporter gene provided a reliable readout of average expression level and expression noise driven by the same $P_{TDH3}$ promoters at the native *TDH3* locus, as measured using Tdh3-YFP fusion proteins (*Figure 2—figure supplement 1A,B*). These fusion protein alleles were not used for comparing fitness effects among *TDH3* promoter alleles because the YFP fusion reduced fitness by 2.5% relative to a strain expressing TDH3 and YFP from independent promoters (*Figure 2—figure supplement 1C*).

A local regression (LOESS) of average expression level on fitness for the 43 *TDH3* alleles and the *TDH3* deletion showed a non-linear relationship with a plateau of maximal fitness near the wild type expression level (*Figure 2C*) similar to that described in *Duveau et al. (2017a)*. Deletion of *TDH3* (expression level of 0% in *Figure 2C*) caused a statistically significant decrease in fitness of 6.1% relative to the wild type allele (t-test, p=$6.4\times10^{-10}$). The minimum change in *TDH3* expression level that significantly impacted fitness was a 14.6% decrease in average expression relative to wild type, which reduced fitness by 0.19% (t-test, p=0.00045). Overexpressing *TDH3* up to 175% did not significantly impact growth rate, but the 207% expression level of the strain carrying a duplication of the wild type *TDH3* gene caused a 0.92% reduction in fitness (*Figure 2C*; t-test, p=$1.4\times10^{-7}$). Notably, none of the 42 mutant alleles of *TDH3* conferred a significantly higher fitness than the wild type allele (one-sided t-tests, p>0.05), indicating that the wild type expression level of *TDH3* is near an optimum for growth in the environment assayed. We expect these differences in fitness among genotypes with different *TDH3* promoter alleles to arise primarily from differences in *TDH3* expression; however, differences in pleiotropy among promoter alleles might also contribute to differences in fitness.

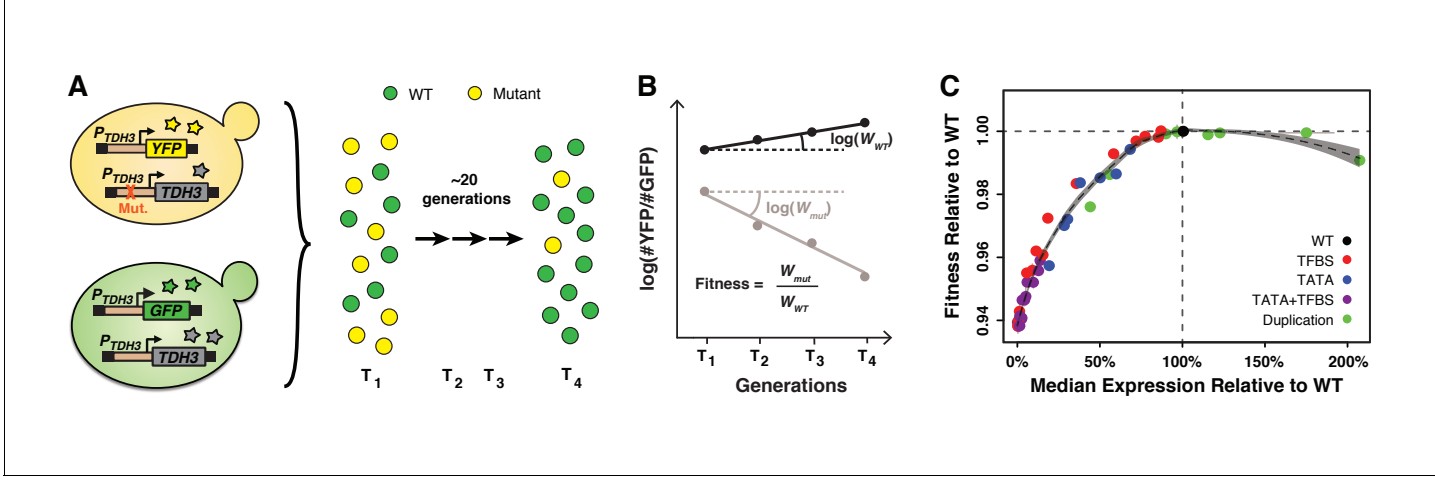

**Figure 2.** Fitness consequences of variation in *TDH3* expression level. (**A**) Overview of competition assays used to quantify fitness. The 43 $P_{TDH3}$ alleles whose activity was described in *Figure 1D* were introduced upstream of the native *TDH3* coding sequence in a genetic background expressing YFP under control of the wild type *TDH3* promoter. A minimum of six replicate populations of the 43 strains were competed for ~20 generations in rich medium (YPD) against a common reference strain expressing GFP under control of the wild type *TDH3* promoter. The relative frequency of cells expressing YFP or GFP was measured every ~7 generations using flow cytometry. (**B**) Competitive fitness was calculated from the change in genotype frequency over time. The relative fitness of each strain was calculated as the mean competitive fitness of that strain across replicates divided by the mean competitive fitness of the strain carrying the wild type allele of *TDH3*. (**C**) Relationship between median expression level of *TDH3* and fitness in rich medium (YPD). Dots show the average median expression and average relative fitness measured among at least four replicates for each of the 43 $P_{TDH3}$ alleles. Colors represent different types of promoter mutation. Error bars are 95% confidence intervals and are only visible when larger than dots. The dotted curve is the best fit of a LOESS regression of fitness on median expression, using a value of 2/3 for the parameter $\alpha$ controlling the degree of smoothing. The shaded area shows the 99% confidence interval of the LOESS fit. Data are available in *Source data 1*. Panels A and B were originally published as *Figure 2A* in *Duveau et al. (2017a)* and are reproduced here by permission of Oxford University Press [http://global.oup.com/academic].
DOI: https://doi.org/10.7554/eLife.37272.007

The following source data and figure supplements are available for figure 2:

**Figure supplement 1.** Comparing effects of 20 alleles of the *TDH3* promoter on expression of the YFP reporter at *HO* and of expression of the TDH3-YFP fusion at the native *TDH3* locus.
DOI: https://doi.org/10.7554/eLife.37272.008

**Figure supplement 1—source data 1.** Activity of 20 $P_{TDH3}$ alleles driving expression of *YFP* at the *HO* locus compared to the activity of the same alleles driving expression of a *TDH3-YFP* gene fusion at the *TDH3* locus.
DOI: https://doi.org/10.7554/eLife.37272.009

**Figure supplement 2.** No significant impact of the genetic background on the expression of the fluorescent reporter.
DOI: https://doi.org/10.7554/eLife.37272.010

## Disentangling the effects of *TDH3* expression level and expression noise on fitness

Residual variation was observed around the LOESS fitted line relating expression level to fitness (*Figure 2C*) that we hypothesized might be explained by differences in noise among genotypes. To examine the effects of differences in expression noise on fitness independent of differences in average expression level, we used the residuals from a local regression of expression noise on expression level for the alleles with average expression levels between 0% and 125% of the wild type allele to define a metric called ΔNoise (*Figure 3A*; *Figure 3—figure supplement 1A–D*). This metric was not significantly correlated with expression level (*Figure 3—figure supplement 2*). *TDH3* alleles with positive ΔNoise values had a higher level of noise than expected based on their expression level and were classified as 'high noise', whereas *TDH3* alleles with negative ΔNoise values had lower levels of noise than expected given their expression level and were classified as 'low noise'.

We then compared the relationship between expression level and fitness for genotypes in the high noise and low noise classes (*Figure 3B*). We found that promoter alleles with positive ΔNoise tended to show a higher fitness than strains with negative ΔNoise (*Figure 3B*, *Figure 3—figure supplement 1E–H*). This beneficial effect of noise on fitness was surprising given prior evidence that selection favored alleles of $P_{TDH3}$ with low expression noise in natural populations (*Metzger et al.,*

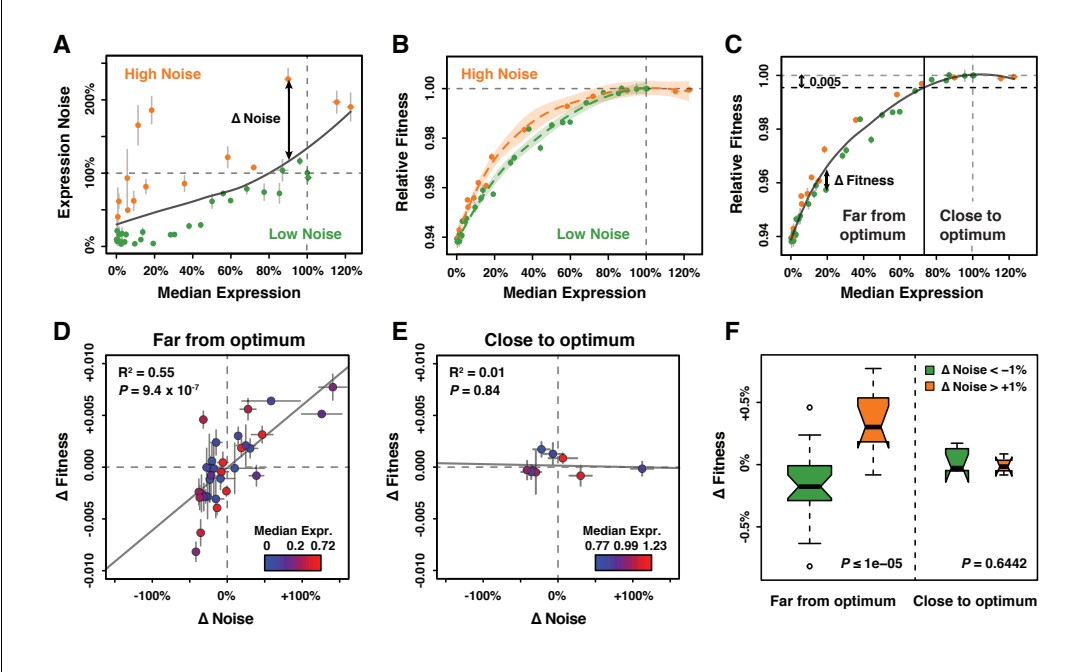

**Figure 3.** Effect of *TDH3* expression noise on fitness. (A) Separation of the 43 $P_{TDH3}$ alleles into two categories based on their effects on median expression level and expression noise (noise strength). The gray curve shows the LOESS regression of noise on median expression using a value of 2/3 for the smoothing parameter. Data points falling below the curve (green) correspond to $P_{TDH3}$ alleles with low noise given their median level of activity. Data points above the curve (orange) correspond to $P_{TDH3}$ alleles with high noise given their median activity. The residual of the LOESS regression ('Δ Noise') is a measure of noise independent of median expression. (B) Relationships between median expression level and fitness for strains with low noise (green, Δ Noise < −1%) and high noise (orange, Δ Noise >+1%). The two LOESS regressions were performed with smoothing parameter $\alpha$ equal to 2/3. (C) Partition of $P_{TDH3}$ alleles into two groups based on the distance of their median activity to the optimal level of *TDH3* expression. The expression optimum (vertical gray dotted line) corresponds to the expression level predicted to maximize fitness from the LOESS regression of fitness on median expression (gray curve). The expression level at which the predicted fitness is 0.005 below the maximal fitness was chosen as the threshold (vertical black dotted line) separating promoters with median activity 'close to optimum' from promoters with median activity 'far from optimum'. The residual of the LOESS regression ('Δ Fitness') is a measure of fitness independent of the median *TDH3* expression level. Dots are colored as in (B). (D) Relationship between Δ Noise and Δ Fitness when median expression is far from optimum. (E) Relationship between Δ Noise and Δ Fitness when median expression is close to optimum. (D–E) The best linear fit between Δ Noise and Δ Fitness is shown as a gray line, with the coefficient of determination ('$R^2$') and the significance of the Pearson's correlation coefficient ('*P*') indicated in the upper left of each panel. Dots are colored based on median expression levels of the corresponding $P_{TDH3}$ alleles as indicated by color gradient. (A–E) Error bars show 95% confidence intervals calculated from at least four replicate samples and are only visible when larger than symbols representing data points. (F) Comparison of Δ Fitness between genotypes with low noise strength (green, Δ Noise < −1%) and genotypes with high noise strength (red, Δ Noise >+1%). Thick horizontal lines represent the median Δ Fitness among genotypes and notches display the 95% confidence interval of the median. Bottom and top lines of each box represent 25th and 75th percentiles. Width of boxes is proportional to the square root of the number of genotypes included in each box. Permutation tests were used to assess the significance of the difference in median Δ Fitness between genotypes with low and high noise and *P*-values are shown in lower right corners. For each test, the values of ΔNoise were randomly shuffled among genotypes 100,000 times. The *P* values shown below each plot represent the proportion of permutations for which the absolute difference in median phenotype between genotypes with low and high ΔNoise was greater than the observed absolute difference in median phenotype between genotypes with low and high ΔNoise. Data are available in ***Source data 1***.

DOI: https://doi.org/10.7554/eLife.37272.011

The following figure supplements are available for figure 3:

**Figure supplement 1.** Calculation of ΔNoise and ΔFitness using four different metrics of noise.
DOI: https://doi.org/10.7554/eLife.37272.012
**Figure supplement 2.** Relationship between median expression level and four different metrics of ΔNoise.
DOI: https://doi.org/10.7554/eLife.37272.013
**Figure supplement 3.** Relationship between median expression level and Δ Fitness.
DOI: https://doi.org/10.7554/eLife.37272.014
**Figure supplement 4.** Relationship between ΔNoise and ΔFitness using four different metrics of noise.
DOI: https://doi.org/10.7554/eLife.37272.015

*Figure 3 continued on next page*

*Figure 3 continued*

**Figure supplement 5.** Robustness of the correlation between ΔNoise and ΔFitness to variation in the smoothing parameter of the LOESS regression used to compute ΔNoise.
DOI: https://doi.org/10.7554/eLife.37272.016

**Figure supplement 6.** Robustness of the correlation between ΔNoise and ΔFitness to variation in the smoothing parameter of the LOESS regression used to compute ΔFitness.
DOI: https://doi.org/10.7554/eLife.37272.017

**Figure supplement 7.** Robustness of the correlation between ΔNoise and ΔFitness to variation in the fitness threshold used to classify genotypes as far or close to optimum.
DOI: https://doi.org/10.7554/eLife.37272.018

**Figure supplement 8.** Fitness, median expression and noise of genotypes with ΔNoise above +1% compared to genotypes with ΔNoise below −1%.
DOI: https://doi.org/10.7554/eLife.37272.019

*2015*). We noticed, however, that the fitness benefit of increasing expression noise was limited to a particular range of average expression levels. Specifically, positive ΔNoise was associated with higher fitness only for average expression levels between 2% and 80% of the wild type expression level (*Figure 3B*). Above 80% of expression, no clear difference in fitness was detected between strains with positive and negative ΔNoise (*Figure 3B*). These same trends were also observed for the other metrics of noise (*Figure 3—figure supplement 1E–H*).

Based on these observations and prior theoretical work (*Tănase-Nicola and ten Wolde, 2008*), we hypothesized that the distance between the average expression level of a $P_{TDH3}$ allele and the optimal level of *TDH3* expression influenced how a change in expression noise impacted fitness. To test this hypothesis, the 43 promoter alleles were split into two groups depending on the distance of their average expression level from the optimal expression level of *TDH3*. Using a local regression of fitness on average expression level, we inferred the value of average expression that would confer a fitness reduction of 0.5% from maximal fitness. Promoter alleles for which the median activity was below this threshold were considered to be 'far from optimum' and promoter alleles with median activity above the threshold were considered to be 'close to optimum' (*Figure 3C*). A metric called ΔFitness, corresponding to the residuals of a local regression of fitness on average expression, was calculated to remove the confounding effect of average expression levels on fitness (*Figure 3C*, *Figure 3—figure supplement 1I–L*, *Figure 3—figure supplement 3*). We found that changes in noise (ΔNoise) and changes in fitness (ΔFitness) were positively correlated for genotypes classified as far from the optimum (Pearson correlation coefficient: r = 0.74, p=9.36×10⁻⁷, *Figure 3D*, *Figure 3—figure supplement 4A–D*), but not for genotypes classified as close to the optimum (r = −0.08, p=0.84, *Figure 3E*, *Figure 3—figure supplement 4E–H*). This result was robust to variation in the choice of the smoothing parameter used for the local regression of noise on average expression, the choice of the smoothing parameter used for the local regression of fitness on average expression, and the fitness threshold used to separate strains with expression levels close and far from optimum (*Figure 3—figure supplements 5*, *6* and *7*). We note, however, that the smaller number of genotypes with mean expression close to the optimum provided less power to detect a significant relationship than genotypes with mean expression far from the optimum.

As an alternative way to test for the impact of expression noise on fitness, we compared ΔFitness for genotypes with positive and negative values of ΔNoise. Permutation tests were used to assess the significance of differences in ΔFitness by randomly shuffling values of ΔNoise among genotypes. Consistent with the correlation analyses, genotypes with positive ΔNoise showed a significantly greater median value of ΔFitness than genotypes with negative ΔNoise at expression levels far from optimum (10⁵ permutations, $P_{\Delta Noise} \leq 10^{-5}$; *Figure 3F*). Among genotypes with average expression close to optimum, no significant difference in median ΔFitness was detected between the positive and negative ΔNoise groups (10⁵ permutations, $P_{\Delta Noise}$ = 0.6442) (*Figure 3F*). The same pattern was observed for all metrics of noise and was not driven by differences in average expression levels between the two ΔNoise groups (*Figure 3—figure supplement 8*).

## Direct measurements of the effect of expression noise on relative fitness

The results presented in the preceding section provide strong evidence that variation in *TDH3* expression can directly affect fitness, but the methods used have at least two limitations. First, ΔNoise and ΔFitness values can be influenced by the set of $P_{TDH3}$ alleles included in the analyses since they are regression residuals. Second, comparisons of fitness among $P_{TDH3}$ genotypes rely upon the assumption that fitness effects are transitive, *i.e.* that differences in fitness between two strains are accurately reported by competitive growth against a third reference strain. Even though such transitivity has often been verified (*de Visser and Lenski, 2002*; *Elena and Lenski, 2003*; *Gallet et al., 2012*), intransitive competition has been observed in several organisms, including yeast (*Paquin and Adams, 1983*). To test whether differences in *TDH3* expression noise affect fitness without calculating regression residuals and without assuming transitivity, we performed direct competition assays between strains with $P_{TDH3}$ promoter alleles that showed similar average expression levels but different levels of expression noise.

Five pairs of *TDH3* alleles for which (i) the median level of activity was similar between the two promoters of each pair, (ii) the level of noise was different between the two promoters of each pair, and (iii) the median level of activity varied among different pairs were chosen from the full set of 171 alleles described above (*Figure 4A*; *Figure 4—source data 1*). The promoter variants of four of these pairs were included in the indirect competition assays and showed the general pattern of increased fitness with increased expression noise when average expression was far from optimum and no significant difference in fitness despite differences in expression noise when average expression was close to optimum (*Figure 4B*). Promoters in the fifth pair were not among the 43 alleles included in the indirect competition experiment but were selected for the direct competition assays because they showed variation in expression noise at average expression levels close to wild type (purple points in *Figure 4A*).

For each of the five pairs, the low noise genotype and the high noise genotype were directly competed against each other under the same conditions used in the competitive growth fitness assay described above except that we doubled the number of generations and the number of replicates to increase the sensitivity of our fitness estimates. In addition, we used pyrosequencing (*Neve et al., 2002*) instead of flow cytometry to determine the relative frequency of the two genotypes at each time point because the two strains competed against each other could not be distinguished based on fluorescence. Relative fitness of the high and low noise genotypes in each pair was calculated based on the changes in relative allele frequency during competitive growth.

For the three pairs of genotypes with an average expression level far from optimum (12%, 19%, and 59% average expression relative to wild type), fitness of the high noise genotype relative to the low noise genotype was significantly greater than 1 (*Figure 4C*), indicating that the high noise genotype grew faster than the low noise genotype. This result was consistent with the differences in fitness measured from the indirect competition assays (*Figure 4B*). By contrast, both pairs of strains with an average expression level closer to the fitness optimum (93% and 102% relative to wild type expression levels) showed slightly but significantly lower fitness of the high noise genotype than the low noise genotype (*Figure 4C*). In these cases, higher expression noise resulted in a ~ 0.1% decrease in fitness relative to lower noise. This difference was detectable with the direct competition assay because the average span of the 95% confidence intervals of fitness estimates was 0.1%, which is half of the 0.2% average 95% confidence intervals from the indirect competition assay described above.

Taken together, our empirical measures of relative fitness show that higher expression noise for *TDH3* is beneficial when average expression level is far from the optimum, but deleterious when average expression is close to the optimum. An intuitive explanation of this phenomenon is that when the average expression level is close to the optimum, increasing expression noise can result in enough cells with suboptimal expression to decrease fitness of the population. Conversely, when the average expression level is far from the optimum, increasing expression noise can result in enough cells with expression closer to the optimum to increase fitness of the population. These effects of expression noise on population fitness can result from differences in expression level among cells causing differences in the cell division rate (a.k.a. doubling time) among cells (*Kiviet et al., 2014*). To better understand the interplay among average expression level, expression noise, and fitness,

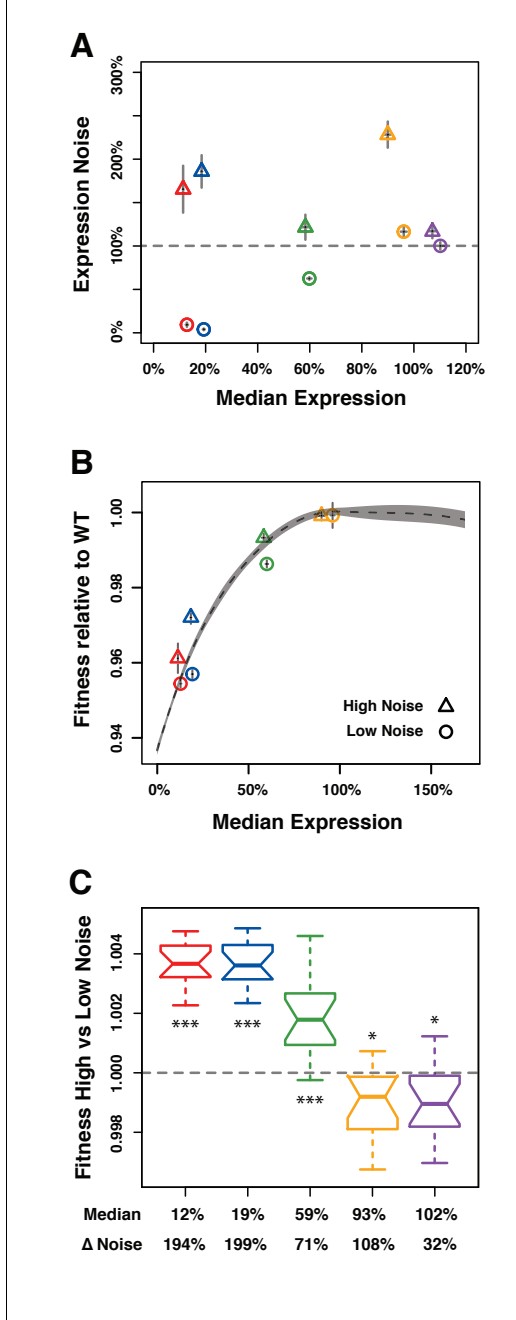

**Figure 4.** Direct competition between genotypes with different levels of noise but similar median levels of *TDH3* expression in glucose. (**A–C**) Different colors are used to distinguish pairs of genotypes ($P_{TDH3}$ alleles) with different median expression levels. (**A**) Median expression level and expression noise (noise strength) for five pairs of genotypes that were competed against each other. Each pair comprises one genotype with low expression noise (circle) and one genotype with high expression noise (triangle). (**B**) Relative fitness for four pairs of genotypes measured in competition assays against the common GFP reference strain. One pair is missing (purple in (**A**)) because the corresponding *Figure 4 continued on next page*

we developed a simple computational model that allowed us to (1) vary the expression mean and noise independently while holding all other parameters constant, (2) track the resulting single cell growth dynamics, and (3) evaluate the consequences for population fitness.

## Simulating population growth reveals fitness effects of noise

To further investigate how the distribution of expression levels among genetically identical cells influences population fitness, we modeled the growth of clonal cell populations that differed in the mean expression level and expression noise for a single gene. In this model, we specified a function defining the relationship between the expression level of a cell and the doubling time of that cell. Following each cell division, the expression levels of mother and daughter cells were sampled independently from an expression distribution characterized by its mean and noise (*Figure 5A*). This independent sampling ignores any inheritance of expression noise, which is a conservative choice for detecting differences in fitness among genotypes due to differences in noise. The doubling time of each cell was then calculated from its expression level (*Figure 5B*), and each clonal population was allowed to expand for the same amount of time, increasing in size at a rate determined by the doubling times of the cells sampled (*Figure 5C*). Empirical measures of single-cell division rates were consistent with these elements of the model, showing more variable cell division times in genotypes with greater *TDH3* expression noise and shorter cell division times in genotypes with mean *TDH3* expression closer to the fitness optimum (*Figure 5—figure supplement 1*). Competitive fitness was ultimately determined in the model by comparing the population size obtained at the end of each simulation experiment to the population size obtain for a constant 'wild type' competitor (*Figure 5D*, *Figure 6—source data 1*). 100 independent simulations were performed for each unique combination of mean expression level and expression noise. Three metrics of expression noise were used for this work: noise strength (similar to Fano factor, *Figure 6*), standard deviation (*Figure 6—figure supplement 1A,C*) and coefficient of variation (*Figure 6—figure supplement 1B,D*).

To calculate doubling times from single cell expression levels, we first used a linear function akin to directional selection in which increases in expression level resulted in shorter doubling times (faster growth) (*Figure 6A*). With this

*Figure 4 continued*

$P_{TDH3}$ alleles were not part of the 43 alleles included in the initial competition experiment. (**A–B**) Error bars show 95% confidence intervals obtained from at least three replicates. (**C**) Competitive fitness of high noise strains relative to low noise strains measured from direct competition assays. Each box represents fitness data from 16 replicate samples. The average median expression level of the two genotypes compared is shown below each box along with the difference in expression noise between these two genotypes (ΔNoise). Thick horizontal lines represent the median fitness across replicates and notches display the 95% confidence interval of the median. The bottom and top lines of each box represent 25th and 75th percentiles. Statistical difference from a fitness of 1 (same fitness between the two genotypes) was determined using *t*-tests (*: $0.01 < P < 0.05$; **: $0.001 < P < 0.01$; ***: $p<0.001$). Data are available in *Figure 4—source data 1*.

DOI: https://doi.org/10.7554/eLife.37272.020

The following source data and figure supplement are available for figure 4:

**Source data 1.** Fitness measured in direct competition assays between strains with low and high values of expression noise.

DOI: https://doi.org/10.7554/eLife.37272.022

**Figure supplement 1.** Median expression level and expression noise for five pairs of genotypes that were competed directly against each other.

DOI: https://doi.org/10.7554/eLife.37272.021

relationship, higher levels of expression noise conferred higher population fitness for a given mean expression level (*Figure 6B*), a pattern more pronounced for high values of mean expression and observed for all metrics of noise (*Figure 6—figure supplement 1A,B*). This finding is consistent with prior work demonstrating that an increased variability of doubling time among individual cells is sufficient to increase fitness at the population level (*Tănase-Nicola and ten Wolde, 2008*; *Cerulus et al., 2016*; *Hashimoto et al., 2016*; *Nozoe et al., 2017*). This is because the doubling time of a population tends to be dominated by the doubling time of the fastest dividing cells in the population, *i.e.* population doubling time is higher than the mean doubling time among all cells in the population.

Next, we used a Gaussian function akin to stabilizing selection in which an intermediate expression level produced the shortest doubling time (faster growth), while lower or higher expression than this optimum would increase doubling time (slower growth) (*Figure 6C*). With this function, we found that the fitness effects of increasing expression noise depended on the mean expression level. Specifically, increasing expression noise increased fitness when the average expression level was far from the optimal expression level and it decreased fitness when the average expression level was close to the optimum (*Figure 6D*), similar to the pattern we observed with our empirical fitness data and in agreement with theoretical work by *Tănase-Nicola and ten Wolde (2008)*. This result was observed for all three metrics of noise, suggesting it is robust to the different scaling relationships between the mean expression level and variability around the mean captured by different metrics of noise (*Figure 6—figure supplement 1C,D*).

These in silico analyses not only provide a plausible mechanistic explanation for our empirical finding that increasing noise can be both beneficial and deleterious in a single environment but they also show that increasing expression noise can alter the effects of changes in mean expression level on fitness. Specifically, when expression noise is high (red lines on *Figure 6D* and *Figure 6—figure supplement 1C,D*), changes in mean expression level are predicted to have much smaller impacts on fitness than equivalent changes when expression noise is low (blue lines on *Figure 6* and *Figure 6—figure supplement 1C,D*). This pattern is also readily apparent when changes in expression noise, instead of changes in mean expression level, are plotted as a function of population fitness (*Figure 6—figure supplement 2*). These observations are consistent with a previously published population genetic model showing that increasing expression noise can reduce the efficacy of natural selection acting on mean expression level (*Wang and Zhang, 2011*).

## Conclusions

Despite many studies providing evidence that natural selection can (*Tănase-Nicola and ten Wolde, 2008*; *Wang and Zhang, 2011*; *Barroso et al., 2018*) and has (*Fraser et al., 2004*; *Lehner, 2008*; *Zhang et al., 2009*; *Metzger et al., 2015*) acted on expression noise, the precise effects of expression noise on fitness have proven difficult to measure empirically. This difficulty arises from the facts that (1) most mutations that alter expression noise also alter mean expression in a correlated fashion, making it difficult to isolate the effects of changes in expression noise on fitness (*Hornung et al.,*

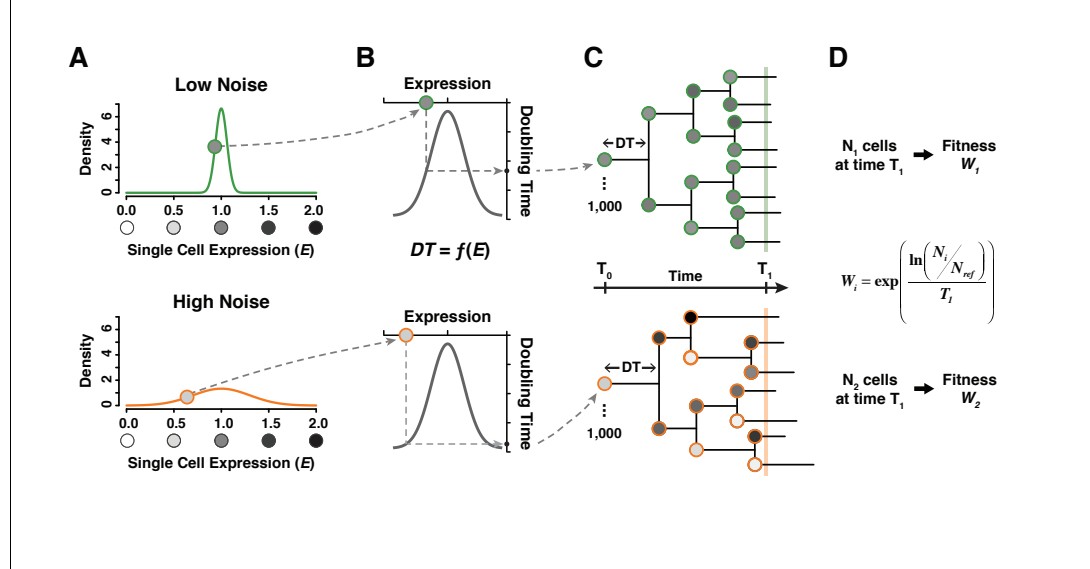

**Figure 5.** A simple model linking single cell expression levels to population fitness. (A) In our model, the expression level $E$ of individual cells is randomly drawn from a normal distribution $N(\mu_E, \sigma_E^2)$. $\sigma_E$ is lower for a genotype with low expression noise (top, green line) and higher for a genotype with high expression noise (bottom, orange line). (B) The doubling time $DT$ of individual cells is directly determined from their expression level using a function $DT = f(E)$. (C) The growth of a cell population is simulated by drawing new values of expression converted into doubling time after each cell division. In this example, doubling time is more variable among cells for the population showing the highest level of expression noise. (D) Population growth is stopped after a certain amount of time (1000 minutes in our simulations) and competitive fitness is calculated from the total number of cells produced by the tested genotype relative to the number of cells in a reference genotype with $\mu_E = 1$ and $\sigma_E = 0.1$. In this example, fitness is lower for the genotype with higher expression noise (bottom) because it produced less cells than the genotype with lower expression noise (top).

DOI: https://doi.org/10.7554/eLife.37272.023

The following source data and figure supplements are available for figure 5:

**Figure supplement 1.** Single-cell division rates estimated using time-lapse microscopy.

DOI: https://doi.org/10.7554/eLife.37272.024

**Figure supplement 1—source data 1.** Single-cell measures of doubling time in four strains with different median levels and noise of TDH3 expression.

DOI: https://doi.org/10.7554/eLife.37272.025

**Figure supplement 1—source data 2.** Summary statistics for comparing the distributions of single-cell doubling time between genotypes with different expression noise levels.

DOI: https://doi.org/10.7554/eLife.37272.026

*2012*; *Keren et al., 2016*; *Liu et al., 2016*), and (2) the magnitude of fitness effects resulting from changes in expression noise is expected to be smaller than that resulting from changes in mean expression level (*Zhang et al., 2009*). In this study, we overcame these challenges by surveying a broad range of mutant promoter alleles for their effects on mean expression level and expression noise, measuring the fitness effects of a subset of these alleles with reduced dependency between effects on mean expression level and expression noise, and using an assay for fitness with power to detect changes as small as 0.1%. We found that the fitness effects of changes in expression noise are indeed generally much smaller than changes in expression level, although they are large enough to be acted on by natural selection in wild populations of *S. cerevisiae* (*Wagner, 2005*; *Metzger et al., 2015*).

We also show that changes in expression noise can be beneficial or deleterious depending on the distance between the mean expression level and the expression level conferring optimal fitness in the environment examined, with increases in expression noise deleterious near the optimal expression level, consistent with data for *TDH3* in *Metzger et al. (2015)*. Although our empirical work focused solely on the *TDH3* gene, the small number of parameters in our simulation model producing the same pattern as these empirical data suggests that the observed relationship among fitness, average expression level and expression noise are likely generalizable to other genes. That said, the precise relationship between expression noise and fitness at the population level is expected to be

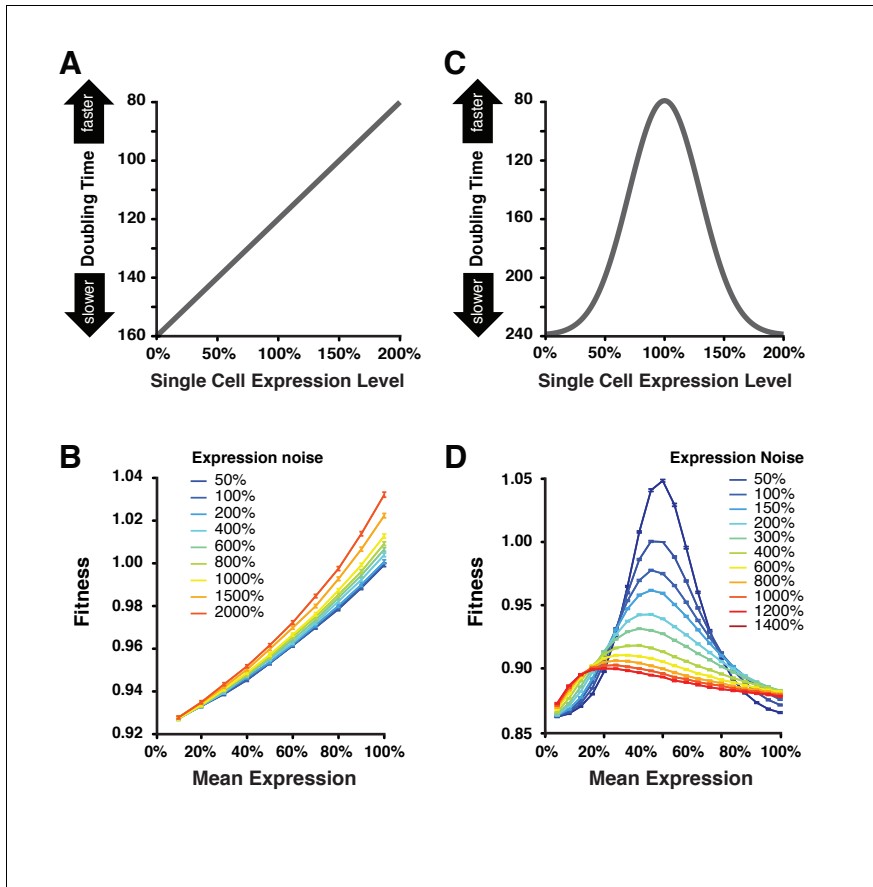

**Figure 6.** Simulating the effect of expression noise on fitness at different median expression levels. (A) The linear function $DT = -40 \times E + 160$ relating the expression level of single cells to their doubling time used for the first set of simulations. (B) Relationship between mean expression ($\mu_E$) and fitness at nine values of expression noise (noise strength: $\sigma_E^2 / \mu_E$) ranging from 50% to 2000% using the linear function shown in (A). (C) Gaussian function $DT = -160 \times \exp\left(-(E-1)^2/0.18\right) + 240$ relating the expression level of single cells to their doubling time used in the second set of simulations. This function shows an optimal expression level at $E = 1$, where doubling time is minimal (*i.e.*, fastest growth rate). (D) Relationship between mean expression ($\mu_E$) and fitness at 11 values of expression noise (noise strength: $\sigma_E^2 / \mu_E$) ranging from 50% to 1400% using the Gaussian function shown in (C). (B, D) Error bars show 95% confidence intervals of mean fitness calculated from 100 replicate simulations for each combination of mean expression and expression noise values. Data are available in *Figure 6—source data 1*.

DOI: https://doi.org/10.7554/eLife.37272.027

The following source data and figure supplements are available for figure 6:

**Source data 1.** Fitness data obtained by modeling the growth of cell populations with different levels of mean expression and expression noise.

DOI: https://doi.org/10.7554/eLife.37272.030

**Figure supplement 1.** Simulating the effect of two different metrics of expression noise on fitness at different median expression levels.

DOI: https://doi.org/10.7554/eLife.37272.028

**Figure supplement 2.** Relationship between expression noise and fitness at different values of mean expression in simulations using a Gaussian function relating single cell expression to doubling time.

DOI: https://doi.org/10.7554/eLife.37272.029

shaped by the relationship between average expression level and doubling time of single cells as well as the temporal dynamics of expression in single cells (*Blake et al., 2006*; *Tănase-Nicola and ten Wolde, 2008*). We provide some experimental measures of single-cell division rates here (*Figure 5—figure supplement 1*), but studies that more directly compare expression levels and division times in individual cells are needed to fully address this issue.

Assuming that the average expression level of a population is near the fitness optimum in a stable environment, but further from the optimum following a change in the environment, our results unify studies showing that increasing expression noise tends to be deleterious in a constant environment but beneficial in a fluctuating one (*Fraser et al., 2004*; *Blake et al., 2006*; *Batada and Hurst, 2007*; *Lehner, 2008*; *Tănase-Nicola and ten Wolde, 2008*; *Zhang et al., 2009*; *Fraser and Kaern, 2009*; *Ito et al., 2009*; *Wang and Zhang, 2011*; *Levy et al., 2012*; *Wolf et al., 2015*; *Liu et al., 2015*; *Keren et al., 2016*). Expression noise may be particularly important in the early phase of adaptation to a fluctuating environment, when a new expression optimum makes an increase in noise beneficial and before expression plasticity evolves as a more optimal strategy (*Wolf et al., 2015*). Such plasticity in expression level seems to have already evolved for *TDH3* (*Duveau et al., 2017b*). Our data suggest that high levels of expression noise can also be beneficial in a stable environment when the mean expression level is far from optimal. For example, if an allele driving suboptimally low expression were to be fixed in a population, selection should initially favor alleles that increase mean expression and/or expression noise. After alleles driving mean expression close to the optimum are fixed, selection should then favor alleles with lower levels of expression noise. The relative frequency by which evolution proceeds through these two paths will depend on both the relative frequency of alleles that increase mean expression and expression noise, as well as the fitness differences between these alleles. We note, however, that the often correlated effects of promoter mutations on mean expression level and expression noise (*Hornung et al., 2012*; *Carey et al., 2013*; *Sharon et al., 2014*; *Vallania et al., 2014*) may limit the ability of natural selection to optimize both mean expression level and expression noise. Future work investigating the effects of other types of mutations on mean expression level, expression noise, and fitness in multiple environments is needed to more fully define the range of variation affecting gene expression upon which natural selection can act.

## Materials and methods

### Yeast strains: genetic backgrounds

All strains used in this work were haploids with similar genetic backgrounds that were derived from crosses between BY4724, BY4722, BY4730 and BY4742 (*Brachmann et al., 1998*) and carry the alleles *RME1(ins-308A); TAO3(1493Q)* from *Deutschbauer and Davis (2005)* and *SAL1; CAT5(91M); MIP1(661T)* from *Dimitrov et al. (2009)* that contribute to increased sporulation efficiency and decreased petite frequency relative to the alleles of the laboratory S288c strain. The construction of this genetic background is described in more detail in *Metzger et al. (2016)*. Strains used to assay transcriptional activity and fitness (described in detail below) had different mating types and drug resistance markers, but these differences did not significantly affect $P_{TDH3}$ transcriptional activity (*Figure 2—figure supplement 2A,B*).

### Yeast strains: construction of strains used to assay transcriptional activity

Transcriptional activity (average expression level and expression noise) was assayed for 171 $P_{TDH3}$ alleles in *S. cerevisiae* strains carrying a fluorescent reporter construct inserted at the *HO* locus on chromosome IV in $MAT\alpha$ cells (*Metzger et al., 2016*). From these alleles (*Figure 1—figure supplements 1—source data 1*), 43 were selected for assaying fitness effects of changing *TDH3* expression (*Source data 1*). 36 of the final 43 $P_{TDH3}$ alleles carried a single copy of a reporter construct consisting of the *TDH3* promoter followed by the *Venus YFP* coding sequence, the *CYC1* terminator and an independently transcribed *KanMX4* drug resistance cassette *Metzger et al. (2016)*. 7 of the final 43 $P_{TDH3}$ alleles variants consist of two copies of the $P_{TDH3}$-YFP-$T_{CYC1}$ construct in tandem separated by a *URA3* cassette. The different $P_{TDH3}$ alleles contain mutations located either in the known binding sites for *GCR1* and *RAP1* transcription factors, in the TATA box or in combinations of both, as described below. The wild type allele of $P_{TDH3}$ consists of the 678 bp sequence located upstream of the *TDH3* start codon in the genome of the laboratory strain S288c, with a single nucleotide substitution that occurred during the construction of the $P_{TDH3}$-YFP-$T_{CYC1}$ construct (A - > G located 293 bp upstream of the start codon). This substitution is present in all $P_{TDH3}$ alleles used in this study.

The effect of this mutation on $P_{TDH3}$ activity in YPD medium was previously described (**Metzger et al., 2015**).

## Single TFBS mutants

A set of 236 point mutations corresponding to almost all C - > T and G - > A substitutions in the *TDH3* promoter was previously inserted upstream of a *YFP* reporter gene on chromosome I in the BY4724 genetic background (**Metzger et al., 2015**). From these, we selected seven $P_{TDH3}$ alleles for which the transcriptional activity spanned a broad range of median fluorescence levels when cells were grown in glucose medium (25% to 90% relative to wild type expression level). These seven promoters carried mutations either in the GCR1 or RAP1 transcription factor binding sites (TFBS) previously characterized in the *TDH3* promoter (**Yagi et al., 1994**). Each $P_{TDH3}$ allele was inserted upstream of *YFP* at the *HO* locus using the *dellitto perfetto* approach (**Stuckey et al., 2011**). Briefly, in the reference strain YPW1002 carrying the wild-type $P_{TDH3}$-*YFP*-$T_{CYC1}$ construct at *HO* (**Metzger et al., 2016**), we replaced $P_{TDH3}$ with a *CORE-UH* cassette (*COunterselectable REporter URA3-HphMX4* amplified from plasmid *pCORE-UH* using oligonucleotides 1951 and 1926 in **Supplementary file 2**) to create strain YPW1784. Then, each of the seven $P_{TDH3}$ alleles was amplified by PCR using oligonucleotides 2276 and 2277 (**Supplementary file 2**) and transformed into YPW1784 to replace the *CORE-UH* cassette and allow expression of *YFP* (**Metzger et al., 2015**). Correct insertion of $P_{TDH3}$ alleles was verified by Sanger sequencing of PCR amplicons obtained with primers 2425 and 1208 (**Supplementary file 2**).

## Double TFBS mutants

To sample average expression levels less than 25% of wild type, we created and measured the activity of 12 $P_{TDH3}$ alleles containing mutations in two different TFBS. We then selected seven of these alleles to be included in the final set of 43 $P_{TDH3}$ alleles (**Source data 1**). Point mutations from different alleles were combined on the same DNA fragment using PCR SOEing (Splicing by Overlap Extension). First, left fragments of $P_{TDH3}$ were amplified from genomic DNA of strains carrying the most upstream TFBS mutations. These PCRs used a common forward primer (2425 in **Supplementary file 2**) and a reverse primer containing the most downstream TFBS mutation to be inserted (P4E8, P4E5, P4G8 or P4G7 in **Supplementary file 2**). In parallel, right fragments of $P_{TDH3}$ were amplified from YPW1002 gDNA using forward primers containing the most downstream TFBS mutations (P1E8, P1E5, P1G8 or P1G7 in **Supplementary file 2**) and a common reverse primer (104 in **Supplementary file 2**). Then, equimolar amounts of the overlapping upstream and downstream fragments of $P_{TDH3}$ were mixed and 25 PCR cycles were performed to fuse both fragments together and to reconstitute the full promoter. Finally, the fused fragments were further amplified for 35 cycles using oligonucleotides 2425 and 1305 (**Supplementary file 2**) and the final products were transformed in strain YPW1784. The presence of desired mutations in $P_{TDH3}$ was confirmed by Sanger sequencing of amplicons obtained with primers 1891 and 1208 (**Supplementary file 2**).

## GCN4 binding sites

To try to create variation in noise independent of the median expression level, we inserted GCN4 binding sites at several locations in the *TDH3* promoter because GCN4 binding sites in synthetic promoters were shown to increase expression noise ($CV^2$) relative to average expression level (**Sharon et al., 2014**). We introduced substitutions in $P_{TDH3}$ to create the GCN4 binding motif TGACTCA at 10 different locations (−121,−152, −184,−253, −270,−284, −323,−371, −407 and −495 upstream of start codon) that originally differed by one, two or three nucleotides from this motif. Targeted mutagenesis was performed using the same PCR SOEing approach as described in **Metzger et al. (2015)** (see **Supplementary file 2** for the list of primers used to insert GCN4 binding sites) and the resulting PCR products were transformed into strain YPW1784. Correct insertion of the TGACTCA motif was confirmed by Sanger sequencing. However, none of the 10 alleles of $P_{TDH3}$ with GCN4 binding sites showed the expected increase in expression noise when cells were grown in glucose (**Figure 1—figure supplement 1**). This could be due to the genomic context being different from the synthetic library used in **Sharon et al. (2014)** or to the fact that $P_{TDH3}$ is one of the most highly active promoters in the yeast genome. None of these 10 alleles were included in the set of 43 $P_{TDH3}$ alleles used for fitness assays.

## TATA box mutants

A second strategy we employed to create variation in expression noise independent of median expression was to mutate the TATA box in the *TDH3* promoter because the presence of a canonical TATA box in yeast promoters has been associated with elevated expression noise (*Newman et al., 2006*). Mutations in the TATA box were also shown to have a clearly distinct impact on expression noise compared to other types of *cis*-regulatory mutations (*Blake et al., 2006*; *Hornung et al., 2012*). We used a random mutagenesis approach to create a large number of alleles with one or several mutations in the $P_{TDH3}$ TATA box. Variants were obtained using PCR SOEing as described above, except that one of the internal overlapping oligonucleotides (primer 2478, *Supplementary file 2*) used to amplify the downstream fragment of $P_{TDH3}$ contained a degenerate version of the wild type TATA box (TATATAAA at position −141 upstream of start codon). This oligonucleotide was synthesized by Integrated DNA Technologies using hand-mixed nucleotides at the eight bases of the TATA box with a proportion of 73% of the wild type nucleotide and 9% of each of the three alternative nucleotides. At this level of degeneracy,~10% of the DNA molecules should carry no mutation,~25% should carry a single mutation in the TATA box,~35% two mutations,~20% three mutations and ~10% four mutations or more. The degenerate primer was used with oligonucleotide 104 to amplify the right fragment of $P_{TDH3}$, and the overlapping primer 2479 was used with oligo 2425 to amplify the left fragment (*Supplementary file 2*). Then, these fragments were fused and amplified as described above for the TFBS mutants. Six independent transformations of the fragments containing random mutations in the TATA box were performed in strain YPW1784 to obtain a large number of colonies. After growth on selective medium (Synthetic Complete medium with 0.9 g/L 5-FluoroOrotic Acid), 244 colonies selected regardless of their fluorescence level were streaked on fresh plates (again SC +5 FOA medium) and then replica plated on YPD +Hygromycin B (10 g/L Yeast extract, 20 g/L Peptone, 20 g/L Dextrose and 300 mg/L Hygromycin B) for negative screening. 106 of the resulting strains turned out not to be fluorescent, among which the vast majority were resistant to Hygromycin B, suggesting they were false positive transformants. The remaining 138 strains were all fluorescent and sensitive to Hygromycin B, as expected from true positive transformants. We then tried to amplify $P_{TDH3}$ in all 244 strains using oligonucleotides 1891 and 1208 (*Supplementary file 2*) and we only observed a band of correct size after electrophoresis for the 138 fluorescent strains. After Sanger sequencing of the PCR products for the 138 positive strains, the type and frequency of mutations observed in the TATA box were found to be very close to expectation (*Figure 1—figure supplements 1—source data 1*). Average expression level and expression noise were measured for all 138 strains as described below. This set of alleles showed broad variation in average expression level (*Figure 1—figure supplement 1*) and had a lower expression noise than TFBS mutations with comparable average expression levels. We selected seven TATA box variants (*Source data 1*) with expression levels ranging from 20% to 75% of wild type to be included in the final set of 43 $P_{TDH3}$ alleles. One of the random TATA box mutants contained a PCR-induced mutation in the GCR1.1 TFBS and was also included in the final set (Var23 in *Source data 1*).

## TATA box and TFBS mutants

To obtain variation in expression noise at expression levels below 20%, we combined mutations in TFBS with mutations in the TATA box in 12 additional $P_{TDH3}$ alleles (*Figure 1—figure supplements 1—source data 1*). Two TATA box variants with 25% and 50% median fluorescence levels were each combined with six different TFBS variants for which median expression ranged from 4% to 45% relative to wild type. The 12 variants were created by PCR SOEing as described above for the double TFBS mutants, except in this case oligonucleotides 2425 and 2788 were used to amplify the upstream $P_{TDH3}$ fragments and oligonucleotides 2787 and 104 were used to amplify the downstream fragments (*Supplementary file 2*). All 12 variants were transformed in strain YPW1784 and confirmed by Sanger sequencing.

## Double-copy constructs

To create variation in average expression level and expression noise for expression levels more than 75% of wild type, we constructed 13 alleles with two copies of the whole $P_{TDH3}$-YFP-$T_{CYC1}$ construct inserted in tandem at the *HO* locus (*Figure 1—figure supplements 1—source data 1*). One of these

constructs carried two copies of the wild type *TDH3* promoter, while the others carried mutated versions of $P_{TDH3}$. We reasoned that the presence of a second copy of the construct would lead to over-expression of YFP, as shown previously (*Kafri et al., 2016*), while differences in noise between the different alleles should be conserved. To construct these alleles, we first fused the selectable marker *URA3* upstream of the $P_{TDH3}$-YFP-$T_{CYC1}$ allele located at the right end of each of the final constructs ('*CONSTRUCT.2*' in *Source data 1*) using PCR SOEing. *URA3* was amplified from the *pCORE-UH* plasmid using oligonucleotides 2688 and 2686 and the 13 $P_{TDH3}$-YFP-$T_{CYC1}$ constructs were amplified from the strains carrying the corresponding $P_{TDH3}$ alleles using oligonucleotides 2687 and 1893 (*Supplementary file 2*). *URA3* and $P_{TDH3}$-YFP-$T_{CYC1}$ were then fused by overlap extension and the resulting fragments were amplified with oligonucleotides 2684 and 2683 (*Supplementary file 2*). Finally, each of the 13 different *URA3*-$P_{TDH3}$-YFP-$T_{CYC1}$ PCR products was transformed in the strain carrying the desired allele of $P_{TDH3}$-YFP-$T_{CYC1}$ (strain carrying '*CONSTRUCT.1*' in *Source data 1*). During these transformations, the *KanMX4* drug resistance cassette was replaced with *URA3*-$P_{TDH3}$-YFP-$T_{CYC1}$ by homologous recombination so that the final constructs were *ho*::$P_{TDH3}$-YFP-$T_{CYC1}$-*URA3*-$P_{TDH3}$-YFP-$T_{CYC1}$. To control for the impact of the URA3 marker on the activity of the *TDH3* promoter, we constructed strain YPW2675 (*ho*::$P_{TDH3}$-YFP-$T_{CYC1}$-*URA3*) by replacing the *KanMX4* cassette with *URA3* amplified using primers 2684 and 2685 (*Supplementary file 2*). YPW2675 was used as the reference when reporting the expression phenotypes (median and noise) of the alleles with two copies of $P_{TDH3}$-YFP-$T_{CYC1}$. To validate the sequence of the full (5.2 kb) constructs, we performed two overlapping PCRs using oligonucleotides 2480 and 1499, and 1872 and 2635 (*Supplementary file 2*). PCR products were sequenced using primers 2480, 1499, 1204, 1872, 2635, 2686, 1305 and 601 in *Supplementary file 2*) to confirm they contained the correct $P_{TDH3}$ alleles. However, using this PCR approach, insertion of more than two tandem copies of $P_{TDH3}$-YFP-$T_{CYC1}$ would remain undetected. Therefore, we used quantitative pyrosequencing to determine the exact number of copies inserted in the 13 strains. We took advantage of the fact that all $P_{TDH3}$ alleles inserted at *HO* carried the mutation A293g upstream of the start codon, while the endogenous *TDH3* promoter did not. This allowed us to determine the total number of $P_{TDH3}$ copies at the *HO* locus by quantifying the relative frequency of A and G nucleotides at position −293 across all copies of the *TDH3* promoter in the genome. For instance, if only one copy of $P_{TDH3}$ is present at *HO*, then the frequency of G at position −293 is expected to be 0.5, since there is one copy of the G allele at the *HO* locus and one copy of the A allele at the endogenous *TDH3* locus. If two copies are present at HO, a frequency of 2/3 is expected for G, and if three copies are present, a frequency of 0.75 is expected. To determine these allele frequencies, we amplified $P_{TDH3}$ in five replicates from all strains carrying two copies of the construct as well as from YPW2675 carrying a single copy using oligonucleotides 2268 and 3094 (*Supplementary file 2*). PCR products were denatured and purified using a PyroMark Q96 Vacuum Workstation (Qiagen) and pyrosequencing was performed on a PyroMark Q96 ID instrument using oligonucleotide 2270 for sequencing (*Supplementary file 2*). Allele frequencies were determined from the relative heights of the peaks corresponding to the A and G alleles on the pyrograms, with the typical correction factor of 0.86 applied to A peaks. Using this method, a small but significant bias toward the G allele was detected, as the observed frequency of G in strain YPW2675 was 0.55 instead of 0.5. This could be caused by PCR bias due to the fact that the A and G alleles are located at different genomic positions. We applied the linear correction y = x * (0.5/0.45)–0.111 to remove the effect of this PCR bias when calculating the frequency of G alleles. Overall, we found that six strains had a frequency of G significantly higher than 2/3 (*t*-test, p<0.05). This suggested that these strains carried more than two copies of the $P_{TDH3}$-YFP-$T_{CYC1}$ construct and they were therefore removed from all subsequent analyses (except Var42 for reasons explained below).

## Extra mutations

Sanger sequencing revealed that a substantial fraction of all $P_{TDH3}$ alleles constructed (~25% of sequenced strains) carried an indel of one nucleotide in one of the homopolymer runs present in the promoter (*Source data 1*). These mutations probably result from polymerase slippage during PCR amplification. For some $P_{TDH3}$ alleles, we were able to isolate independent clones that differed only by the presence or absence of these homopolymer mutations, giving us the opportunity to test the impact of homopolymer length variation on transcriptional activity. Using the fluorescence assay described below, we found that *del431A*, *del54T* and *ins432A* had no detectable effect on median

expression level or expression noise (*Figure 2—figure supplement 2C,D*). Therefore, strains carrying these mutations were included in the expression and fitness assays.

## Yeast strains: construction of strains used to assay fitness

The strains described above all carried the *ho::P$_{TDH3}$-YFP-T$_{CYC1}$* reporter construct, allowing sensitive quantification of the transcriptional activity of different *P$_{TDH3}$* alleles. In these strains, the endogenous promoter driving expression of the native TDH3 protein was left unaltered. To measure how variation in TDH3 protein levels induced by mutations in the *TDH3* promoter could impact cell growth, we inserted the final set of 43 *P$_{TDH3}$* alleles described above upstream of the endogenous *TDH3* coding sequence. *P$_{TDH3}$* variants were integrated in the genetic background of strain YPW1001, which is almost identical to the reference strain YPW1002 used for the expression assays, except that the mating type of YPW1001 is *MAT*a and it carries a *P$_{TDH3}$-YFP-T$_{CYC1}$-NatMX4* construct at *HO* conferring resistance to Nourseothricin. The reporter construct served a dual purpose: it ensured that the strains used in the expression and fitness assays carried the same number of copies of *TDH3* promoter in their genomes and it allowed high-throughput counting of yellow-fluorescent cells carrying *P$_{TDH3}$* variants in the competition experiments described below. Importantly, we did not detect any difference in fluorescence levels between strains YPW1002 and YPW1001 (*Figure 2—figure supplement 2A,B*), indicating that the few genetic differences between the background of the strains used in the expression and fitness assays did not significantly affect the activity of the *TDH3* promoter.

## Single-copy constructs

To insert the 35 alleles containing a single copy of *P$_{TDH3}$* at the native *TDH3* locus, we first replaced the endogenous *TDH3* promoter of strain YPW1001 with a *CORE-UK* cassette (*URA3-KanMX4*) amplified with oligonucleotides 1909 and 1910 (*Supplementary file 2*) to create strain YPW1121. Then, the 35 *P$_{TDH3}$* alleles were amplified from the *HO* locus in the strains previously constructed (*Source data 1*) using oligonucleotides 2425 and 1305 (*Supplementary file 2*). PCR products were purified using a DNA Clean and Concentrator kit (Zymo Research), amplified using primers 1914 and 1900 (*Supplementary file 2*) to attach appropriate homology tails and transformed in strain YPW1121. In addition, because all the *P$_{TDH3}$* variants inserted at *HO* carried the PCR-induced mutation A293g, we created the strain YPW1189 that carried mutation A293g in the endogenous *TDH3* promoter. YPW1189 served as the reference strain when calculating relative fitness. In all these strains, the presence of the correct mutations in *P$_{TDH3}$* at the native locus was confirmed by Sanger sequencing of PCR products obtained with oligonucleotides 1345 and 1342 (*Supplementary file 2*).

## Double-copy constructs

To measure the impact on fitness of overexpression of the native TDH3 protein, we created seven tandem duplications of the whole *TDH3* locus (*TDH3::P$_{TDH3}$-TDH3-URA3- P$_{TDH3}$-TDH3*) that contained the same combinations of promoter alleles as those inserted at *HO* (*Source data 1*). Duplications of *TDH3* were built in a similar way as the double-copy constructs inserted at *HO*. First, *URA3* was amplified from the *pCORE-UH* plasmid using oligonucleotides 2688 and 2686 and the *TDH3* variants corresponding to the copy located on the right in the final constructs ('*CONSTRUCT.2*' in *Source data 1*) were amplified using oligonucleotides 2687 and 1893 (*Supplementary file 2*). *URA3* and *P$_{TDH3}$-TDH3* PCR products were then fused by overlap extension and the resulting fragments were amplified with oligonucleotides 2696 and 2693 (*Supplementary file 2*). Finally, each of the seven different *URA3-P$_{TDH3}$-TDH3* products was transformed in the strain carrying the desired allele for the left *P$_{TDH3}$-TDH3* copy ('*CONSTRUCT.1*' in *Source data 1*). To control for the impact of URA3 expression on fitness, we constructed strain YPW2682 (*TDH3::P$_{TDH3}$-TDH3-URA3*) by transforming a *URA3* cassette amplified from plasmid *pCORE-UH* with oligonucleotides 2696 and 2697 in strain YPW1189. YPW2675 was used as the reference when reporting the relative fitness of the seven strains carrying two copies of *TDH3*. To sequence the full *TDH3* duplications (5.5 kb), we performed four overlapping PCRs using oligonucleotides 1345 and 1499, 2694 and 1911, 2670 and 1342, 601 and 2695 and sequenced them with oligonucleotides 1345, 1499, 601, 2691, 2053, 2670, 1342, 601, 2695 (*Supplementary file 2*).

As described for the double-copy constructs at *HO*, we used quantitative pyrosequencing to determine the exact number of *TDH3* copies inserted in the seven strains. However, we could not directly quantify the frequency of mutation A293g in these strains, because all copies of *TDH3* promoters present in their genomes carry the G mutation. Therefore, we first crossed all seven strains to YPW1139 (*Metzger et al., 2016*), a strain that contains the A allele at position −293 of the native *TDH3* promoter. In the resulting diploids, the frequency of G allele at the native *TDH3* locus is expected to be 0.5 if the original haploid strain carried a single copy of *TDH3* at the native locus, 2/3 if it carried two copies of *TDH3* at the native locus and 3/4 if it carried three copies. To determine allele frequency at position −293 of $P_{TDH3}$ for the native *TDH3* locus only, we amplified the promoter using primers 2268 and 3095 specific to the native locus (*Supplementary file 2*) and then used pyrosequencing as described above. We found that one strain carried three copies of *TDH3* at the native locus instead of two (*Figure 4—figure supplement 1—source data 1*). However, we did not exclude the corresponding variant (Var42) from subsequent analyses, because it also integrated three copies of the reporter construct at *HO*.

Finally, during growth rate assays, cells carrying a tandem duplication of *TDH3* could potentially lose a copy of *TDH3* through intrachromosomal homologous recombination, which could affect fitness estimates. In strains carrying *TDH3::TDH3-URA3-TDH3* constructs, the loss of a *TDH3* copy by recombination should be accompanied by the deletion of the *URA3* marker. To estimate how frequently such recombination events might occur, we quantified the frequency of Ura- cells in strain YPW2679 (*TDH3::TDH3-URA3-TDH3*) at four time points over the course of 50 generations of growth in similar conditions as used in competition growth assays. Four replicate cultures of YPW2679 were grown to saturation in SC - Ura medium at 30°C. Then, 0.1 ml of each culture was plated on SC +5 FOA medium and each culture was diluted to a density of $10^4$ cells/ml in YPD rich medium. Dilution to $10^4$ cells/ml in YPD was repeated every 12 hr for 72 hr and plating on SC +5 FOA was repeated every 24 hr. After three days of incubation at 30°C, colony-forming units were counted on all SC +5 FOA plates, allowing the estimation of the frequency of Ura- cells every ~17 generations for a total of ~50 generations. The frequency of Ura- cells was found to increase during the first 34 generations of growth before reaching a plateau representing a state of mutation-selection balance. At this stage, the average frequency of Ura- cells was about $5.2 \times 10^{-5}$. Therefore, even if spontaneous loss of one *TDH3* copy occurred in a fraction of cells, these events were found to be too rare to have a significant impact on fitness estimates. Data used to estimate the frequency of intrachromosomal recombination can be found in *Supplementary file 1* – Dataset 6.

### *TDH3* deletion

We deleted the native *TDH3* locus in the genetic background of strain YPW1001 to create strain YPW1177. To do this, we amplified a region of 171 bp immediately upstream of the *TDH3* promoter using oligonucleotides 1345 and 1962 (*Supplementary file 2*). Oligonucleotide 1962 is composed of a 5' sequence of 22 nucleotides priming directly upstream of the *TDH3* promoter fused to a 3' sequence of 38 nucleotides homologous to the 3'UTR sequence immediately downstream of *TDH3* coding sequence. Therefore, transformation of the PCR product in strain YPW1121 (*tdh3::URA3-KanMX4-TDH3*) led to the deletion of the *URA3-KanMX4* cassette and of the *TDH3* coding sequence. In this strain, both the *TDH3* promoter and the *TDH3* coding sequence are deleted, and the coding sequence of the upstream gene *PDX1* is fused to the terminator sequence of *TDH3*, so that *PDX1* would remain functional. Correct deletion of *TDH3* was confirmed by Sanger sequencing of the region amplified with oligonucleotides 1345 and 2444 (*Supplementary file 2*) in strain YPW1177.

### GFP competitor

To measure how variation in *TDH3* expression affected growth rate, the strains described above were all grown competitively against a common strain, YPW1160, which carried a $P_{TDH3}$-GFP-$T_{CYC1}$-KanMX4 construct inserted at the *HO* locus in the same genetic background as the other strains. The expression of Green Fluorescent Protein in YPW1160 cells allowed for highly efficient discrimination from cells expressing the Yellow Fluorescent Protein using flow cytometry. To construct strain YPW1160, the GFP-$T_{CYC1}$ sequence was amplified from strain YPW3 (*swh1::$P_{TDH3}$-GFP-$T_{CYC1}$*, obtained from Barry Williams) using oligonucleotides 601 and 2049 (*Supplementary file 2*). In

parallel, *KanMX4* was amplified from strain YPW1002 using oligonucleotides 2050 and 1890 (*Supplementary file 2*). The two fragments were fused by PCR SOEing and the product was amplified using oligonucleotides 601 and 1890 (*Supplementary file 2*) before transformation in strain YPW1001 (*ho::P_{TDH3}-YFP-T_{CYC1}-NatMX4*). Selection on G418 allowed the recovery of cells that switched the *YFP-T_{CYC1}-NatMX4* cassette for the *GFP-T_{CYC1}-KanMX4* cassette. The fluorescence emission detected on the flow cytometer was consistent with expression of GFP.

## Expression assays

### Quantification of fluorescence using flow cytometry

Fluorescence level was measured as a proxy for $P_{TDH3}$ transcriptional activity using flow cytometry as described in (*Metzger et al., 2016*). All strains were revived from −80°C glycerol stocks on YPG plates (10 g Yeast extract, 20 g Peptone, 30 ml Glycerol, 20 g agar per liter) and, after 2 days of growth, arrayed in 96-well plates containing 0.5 ml of YPD medium per well. In addition to the tested strains, the reference strain YPW1002 was inoculated in 24 positions, which were used to correct for plate and position effects on fluorescence. Strain YPW978, which does not contain the *YFP* reporter construct (*Metzger et al., 2016*), was inoculated in one well per plate and used to correct for autofluorescence. Cells were maintained in suspension at 250 rpm by the presence of a 3 mm glass bead in each well. After 20 hr of growth at 30°C, cells were transferred on YPG omnitrays using a V and P Scientific pin tool and grown for 2 days. Next, samples from each omnitray were inoculated in six replicate 96-well plates in 0.5 ml of YPD and grown for 22 hr at 30°C until they reached saturation. At this point, 15 μl of each culture was diluted into 0.5 ml of PBS (phosphate-buffered saline) in 96-well plates. Fluorescence was recorded for ~20,000 events per well using a BD Accuri C6 instrument coupled with a HyperCyt autosampler (IntelliCyt Corp). A 488 nm laser was used for excitation and a 530/30 optical filter for acquisition of the YFP signal. A modified version of this protocol was used to measure the fluorescence of the final set of 43 $P_{TDH3}$ variants with experimental conditions more similar to those experienced during the competition growth assays. After the 22 hr of growth in YPD, samples were not immediately run on the flow cytometer, but instead they were diluted to fresh medium every 12 hr for 36 hr to reach steady exponential phase of growth. Prior to each dilution, cell density was measured for all samples using a Sunrise absorbance reader (Tecan) and one dilution factor was calculated for each 96-well plate so that the average cell density would reach $5 \times 10^6$ cells/ml after 12 hr of growth. This procedure ensured that all samples were maintained in constant exponential growth since no sample reached a density above $10^7$ cells/ml, while limiting the strength of genetic drift since the number of cells transferred during dilution was larger than 10,000. Another difference with the protocol mentioned above is that no glass bead was added to the plates. Instead, cells were maintained in suspension by fitting the culture plates on a rotating wheel. After 36 hr of growth, samples were diluted to $2.5 \times 10^6$ cells/ml in PBS and the fluorescence of 20,000 events per well was acquired by flow cytometry. Flow data (FCS files) used to quantify fluorescence levels produced by the 43 *TDH3* promoter alleles are available in the FlowRepository (flow-repository.org) under experiment ID FR-FCM-ZY8Y.

### Relationship between mRNA levels and fluorescence

The relationship between fluorescence intensity measured by flow cytometry and fluorophore concentration in a cell is expected to be positive and monotonic, but this relationship is not necessarily linear (Wang and Gaigalas 2011). For most flow cytometers, the photomultiplier tube (PMT) voltages can be calibrated to approach a linear relationship for the range of fluorescence intensities covered by the samples, but this cannot be done on the Accuri C6 because PMT voltages are fixed. Instead, we empirically determined the function relating fluorescence intensities to *YFP* mRNA levels using eight strains with different fluorescence levels and then we applied this function to transform fluorescence intensities for each cell of every sample. The function between fluorophore concentration ($y$) and fluorescence intensity ($x$) was previously determined to be of the form $\log(y) = a \times \log(x) + b$ (Wang and Gaigalas 2011). In our case, $y$ represents mRNA level instead of fluorophore concentration, but this should not affect the shape of the function since previous studies found a linear relationship between mRNA levels and fluorophore concentration (Wolf *et al.* 2015; Kafri *et al.* 2016). To determine the constants $a$ and $b$, we measured fluorescence intensity and *YFP mRNA* levels in eight strains covering the whole range of fluorescence levels expressed by the strains included in this

study. First, three replicates of YPW978 (non-fluorescent strain), YPW2683 (ho::$P_{TDH3}$-YFP-$T_{CYC1}$/ho::$P_{TDH3}$-GFP-$T_{CYC1}$ diploid) and seven strains carrying variants of the ho::$P_{TDH3}$-YFP-$T_{CYC1}$ construct with different $P_{TDH3}$ alleles (*Figure 1—source data 1*) were grown for 24 hours at 30°C in 5 ml of YPD, along with 24 replicates of strain YPW1182 expressing GFP (same genetic background as YPW1160, except with *MATα* mating type). All samples were then diluted to a density of 2 x 10$^6$ cells/ml in fresh YPD medium and grown for another 4 hours. Next, 0.1 ml of each culture was transferred to 0.4 ml of PBS and fluorescence intensity was immediately scored for ~20,000 events per sample on the BD Accuri C6 instrument.

In parallel, for each replicate of the tested strains, 0.5 ml of culture was mixed with 0.5 ml of one of the 24 cultures of YPW1182 strain in a microcentrifuge tube. Genomic DNA and RNA were co-extracted from the 24 mixed populations using a modified version of Promega SV Total RNA Isolation System and cDNA was synthesized from RNA samples as previously described (Metzger *et al.* 2015). Then, pyrosequencing was performed to quantify the relative frequency of *YFP* and *GFP* sequences in gDNA and cDNA samples (Wittkopp 2011). The pyrosequencing assay was designed to quantify allele frequency at a position located 607 bp downstream of the *YFP* start codon, for which a GT/TA difference exists between *YFP* and *GFP* coding sequences. A region that encompassed this polymorphism was amplified in all gDNA and cDNA samples using oligonucleotides 2723 and 2725 (*Supplementary file 2*). These oligonucleotides were designed to anneal both to *YFP* and *GFP* coding sequences, which are 98% identical. Pyrosequencing was performed on a Pyro-Mark Q96 ID instrument using oligonucleotide 2726 for sequencing (*Supplementary file 2*). Because the sequenced region contained two variable positions (G/T and T/A), we determined allele frequencies separately for each position and used the average as the relative frequency of *YFP* and *GFP* alleles. For each sample, we then calculated *YFP* mRNA level relative to the reference strain YPW1002 using the measured frequency of *YFP* allele in gDNA and cDNA. First, we corrected for small biases in allele frequencies that could be caused by PCR bias. To do this, we took advantage of the fact that true allele frequencies were known for gDNA samples of YPW1002 (100% *YFP*), YPW2683 (50% *YFP*) and YPW1182 (0% *YFP*). We regressed the measured allele frequencies on the true allele frequencies for all gDNA replicates of these three samples using R *smooth.spline* function. The fitted model was then used to correct allele frequencies for all other gDNA and cDNA samples. Our next goal was to calculate $A$, defined as the abundance of *YFP* mRNA expressed by each tested strain relative to the abundance of *GFP* mRNA expressed by YPW1182. If $G$ is the frequency of *YFP* allele in the gDNA sample and $C$ the frequency of *YFP* allele in the cDNA sample, then $C = \frac{A \times G}{A \times G + 1 \times (1-G)}$. From this equation, we can deduce that $A = \frac{(1-G) \times C}{(1-C) \times G}$. We applied this formula to our measured estimates of $G$ and $C$ to calculate $A$. For each sample, the calculated value of $A$ was divided by the value obtained for the reference strain YPW1002 to obtain an estimate of *YFP* mRNA level expressed relative to the reference. Finally, we identified the function of form $\log(y) = a \times \log(x) + b$ that best fitted to our measures of mRNA levels and fluorescence intensities using R function *nls*. The least-square estimates of the parameters were $a = 10.469$ and $b = -9.586$. As expected, we observed a nonlinear relationship between *YFP* mRNA level and fluorescence intensity (Figure 1B, $R^2$ = 0.83), but a linear relationship between the logarithm of *YFP* mRNA level and the logarithm of fluorescence intensity (Figure 1C, $R^2$ = 0.99). Data used to establish the relationship between YFP mRNA levels and fluorescence can be found in *Figure 1 – source data 1* and *Supplementary file 1* – Dataset 1.

## Analysis of flow cytometry data for expression

Flow cytometry data were analyzed using R packages *flowCore* (Hahne *et al.* 2009) and *flowClust* (Lo *et al.* 2009) with modifications of the methods described in Duveau *et al.* (2017a) linked to the transformation of fluorescence intensities mentioned above. First, the clustering functions of *flowClust* were used to filter out all events that did not correspond to single cells based on the height and the area of the forward scatter signal. Then, the intensity of the fluorescence signal was scaled by cell size in several steps. We first performed a principal component analysis on the logarithm of forward scatter (FSC.A) and logarithm of fluorescence (FL1.A) for all filtered events. Next, we defined the vector $\nu$PTDH3 between the origin and the intersection of the two eigenvectors. We then calculated the angle $\theta$PTDH3 between the first eigenvector and $\nu$PTDH3. FSC.A and FL1.A data were transformed by a rotation of angle $\theta$PTDH3 centered on the intersection between the two eigenvectors.

Finally, for each event, the transformed FL1.A value was divided by the transformed FSC.A value to obtain a measure of fluorescence level independent of cell size. The fluorescence level of each individual cell was then rescaled using the function $\log(y) = 10.469 \times \log(x) - 9.586$PTDH3 to follow a linear relationship with *YFP* mRNA levels, as explained in the previous paragraph. For each sample, the median $m_{YFP}$PTDH3 and the standard deviation $s_{YFP}$PTDH3 of expression were calculated from the fluorescence levels of at least 1000 cells. Next, we corrected for variation in fluorescence levels caused by factors beyond experimental control by using the 24 control samples present on each plate at the same positions. For each environment, $\log(m_{YFP})$PTDH3 and $\log(s_{YFP})/\log(m_{YFP})$PTDH3 of control samples were fitted to a linear model that included explanatory variables such as average cell size, replicate, plate, row, column and flow run. The variable that had the greatest impact on fluorescence was found to be "flow run". This effect is caused by random variation in the sensitivity of the flow cytometer to measure fluorescence intensity between each run of 48 samples, rather than actual variation in YFP expression, as indicated by the observation of random shifts when running the same plate multiple times. Therefore, for each sample, the effect of "flow run" was removed on a scale that was linearly related to fluorescence intensity and not mRNA level. Given that the logarithm of mRNA levels $\log(y)$PTDH3 scales linearly with the logarithm of fluorescence intensity, the linear model to correct for "flow run" and "row" effects was applied on linear estimates of $median(\log(y))$PTDH3 and $\sigma(\log(y))$PTDH3. $\log(m_{YFP})$PTDH3 scales linearly with $median(\log(y))$PTDH3 and $\log(s_{YFP})/\log(m_{YFP})$PTDH3 is expected to scale approximately linearly with $\sigma(\log(y))$PTDH3. Indeed, the delta method (Ver Hoef 2012) postulates that $\sigma^2(f(x)) = \sigma^2 \times (f'(x))^2$PTDH3 and the first derivative of $\log(x)$PTDH3 is $1/x$PTDH3. The corrected values of $\log(m_{YFP})$PTDH3 and $\log(s_{YFP})/\log(m_{YFP})$PTDH3 were then used to calculate corrected values for $m_{YFP}$PTDH3 and $s_{YFP}$PTDH3. This procedure was found to uniformly decrease the variance of $m_{YFP}$PTDH3 and $s_{YFP}$PTDH3 among replicates of a same strain, independently of the average $m_{YFP}$PTDH3 of the strain. Next, we corrected for autofluorescence by subtracting the mean values of $m_{YFP}$PTDH3 and $s_{YFP}$PTDH3 among replicates of the non-fluorescent strain YPW978 from the values of $m_{YFP}$PTDH3 and $s_{YFP}$PTDH3 of each sample. At this stage, in addition to $s_{YFP}$PTDH3, we calculated three other metrics of expression variability (i.e., noise), $CV^* = \frac{s_{YFP}}{m_{YFP}}$PTDH3, $logCV^* = \log_{10}\left(\frac{s_{YFP}}{m_{YFP}}\right)$PTDH3 and *Noise strength* $= \frac{s_{YFP}^2}{m_{YFP}}$PTDH3. These metrics are similar to the coefficient of variation, the logarithm of the coefficient of variation and the Fano factor (Kaern *et al.* 2005), except that $m_{YFP}$PTDH3 is a median instead of a mean. The four metrics of expression noise were used in parallel in all subsequent analyses. For each strain, samples for which $m_{YFP}$PTDH3 and $s_{YFP}$PTDH3 departed from the median value among replicate populations by more than four times the median absolute deviation were discarded. For each sample, median expression and expression noise were then divided by the mean phenotypic value obtained among replicate populations of the reference strain YPW1002 (for single-copy $P_{TDH3}$ variants) or YPW2675 (for double-copy $P_{TDH3}$ variants). Finally, these relative measures of median expression and expression noise were averaged among replicate populations of each genotype. Processed fluorescence data can be found in *Supplementary file 1* – Dataset 2 and *Source data 1*.

## Fitness assays
### Doubling time in each environment
Prior to performing competition assays, we measured the doubling time of the reference strain YPW1160 when grown in YPD medium. Three replicate cultures of YPW1160 were started in parallel in 5 ml of YPD and incubated for 36 hours at 30°C with dilution to $5 \times 10^5$ cells/ml every 12 hours. After the last dilution, cell density $D$ was quantified every 60 minutes for 10 hours and then after another 800 minutes by measuring optical density at 660 nm. Doubling time was calculated as the inverse of the slope of the linear regression of $\log(D)/\log(2)$ on time during the logarithmic phase of growth where the relationship between $\log(D)$ and time is linear. The average doubling time of the reference strain YPW1160 used in subsequent competition assays was found to be 80 minutes in YPD.

## Competition assays against a common reference

Because the deletion of *TDH3* is known to cause only a ~ 5% reduction in growth rate, detecting a significant impact on fitness of a change in *TDH3* expression level or expression noise required highly accurate measurements of growth rate. For this reason, we decided to use head-to-head competition assays between strains expressing different levels of TDH3 protein and a common reference strain (YPW1160) to measure relative growth rate, which is a more sensitive method than directly measuring the absolute growth rate of each isolated strain (*Gallet et al., 2012*). Indeed, the additive effect of micro-environmental variation on growth estimates is nullified during competitive growth, because the two competitors are grown in the same microenvironment. Relative growth rate during log-phase was used as a proxy for fitness in this study, although this is not the only component of fitness.

To identify experimental conditions that would allow accurate estimates of fitness (precision of at least $10^{-3}$) while keeping cost and labor reasonable, we first performed a power analysis based on simulations to determine how experimental parameters affected accuracy. We decided that during the competition assays cells would be maintained in the logarithmic phase of growth by repeated dilutions in small volume of medium in 96-well plates to handle large number of samples in parallel. Six different parameters associated with this experimental design were varied in the simulations: (1) The number of biological replicates for each sample; (2) The starting frequency of the two strains competed against each other; (3) The difference in fitness between the two competitors; (4) The number of generations after dilution to fresh medium, a parameter that determined the number of cells transferred (or the bottleneck size) after each dilution; (5) The number of dilution cycles, which also determined the number of times the relative frequency of the competed strains was assessed; (6) The number of cells counted each time the relative frequency of the competed strains was assessed. For each of 20,160 combinations of parameter values, the competition assay was simulated 5000 times to estimate the standard deviation of the selection coefficient. Then, given this standard deviation and the tested number of replicates, R function *power.t.test* was used to determine the minimum difference in selection coefficient that could be detected with a significance level of 0.05 and a power of 0.95. All six parameters were found to have an impact on the precision of the selection coefficient, but to different degrees. Interestingly, precision was maximized for intermediate values of the number of generations between two consecutive dilutions and for intermediate values of the total number of dilution cycles, because of the impact of these parameters on genetic drift. To achieve a precision close to $10^{-3}$ in the actual competition experiment, we decided to use eight replicates per sample, to mix the two competing strains in equal proportion, to use a common competitor strain (YPW1160) that had a similar fitness as the wild-type strain (YPW1189), to grow cells for about eight generations of the common competitor after each dilution, to use a total of four phases of growth followed by dilutions and to score genotype frequencies at four time points by screening at least 50,000 cells per sample on the flow cytometer.

The tested strains carrying different alleles of the *TDH3* promoter at the native locus and expressing YFP (*Source data 1*) were arrayed on four 96-well plates in 0.5 ml of YPD, with two replicates of each strain on each plate. In parallel, the common competitor YPW1160 expressing GFP was also arrayed on four 96-well plates in 0.5 ml of YPD. All plates were incubated for 24 hr at 30°C on a rotating wheel. After measuring cell densities using a Sunrise plate reader, an equal volume of YFP and GFP cell cultures were mixed together and diluted in 0.5 ml of YPD in four 96-well plates. The dilution factor was calculated for each plate based on the doubling time of the GFP strain (YPW1160) so that the average cell density would reach ~$5 \times 10^{6}$ cells/ml after 12 hr of growth. This procedure of cell density measurement and dilution followed by 12 hr of growth was repeated three times and constituted the acclimation phase of the experiment, during which the relative frequency of YFP and GFP strains was not recorded. After this acclimation phase, samples were diluted every 10 hr in fresh YPD for a total of 30 hr of exponential growth. Cell density was measured for all samples prior to each dilution. Immediately after dilution to fresh medium, samples were diluted in 0.3 ml of PBS to a final density of $1.5 \times 10^{6}$ cells/ml in four 96-well plates and placed on ice to stop growth. ~75,000 events were recorded for each sample on a BD Accuri C6 flow cytometer, using a 488 nm laser for excitation and two different optical filters (510/10 and 585/40) to acquire fluorescence. These filters allowed separation of the GFP and YFP signals. With this protocol, the relative frequency of YFP and GFP cells was measured at four time points during the competition assays.

Flow cytometry data (FCS files) used to quantify competitive fitness for the 43 *TDH3* promoter alleles are available in the FlowRepository (flowrepository.org) under experiment ID FR-FCM-ZYJN (strains with a single copy of *TDH3* at the native locus) and FR-FCM-ZY7E (strains with two copies of *TDH3* at the native locus). For analysis of flow data using R scripts provided in *Supplementary file 3*, FCS files need to be sorted as follows. The 96 files with a name starting by 'EXP.C_T1' in experiment FR-FCM-ZY7E should be inserted between files 'TimePoint1_Plate3_H12.fcs' and 'TimePoint2_Plate1_A01.fcs' in experiment FR-FCM-ZYJN. The 96 files with a name starting by 'EXP.C_T2' in experiment FR-FCM-ZY7E should be inserted between files 'TimePoint2_Plate3_H12.fcs' and 'TimePoint3_Plate1_A01.fcs' in experiment FR-FCM-ZYJN. The 96 files with a name starting by 'EXP.C_T3' in experiment FR-FCM-ZY7E should be inserted between files 'TimePoint3_Plate3_H12.fcs' and 'TimePoint4_Plate1_A01.fcs' in experiment FR-FCM-ZYJN. The 96 files with a name starting by 'EXP.C_T4' in experiment FR-FCM-ZY7E should be inserted after the last file 'TimePoint4_Plate3_H12.fcs' in experiment FR-FCM-ZYJN.

## Analysis of flow cytometry data for fitness

The number of cells expressing either YFP or GFP was counted for each sample using custom R scripts (*Supplementary file 3*). After $\log_{10}$ transformation of the raw data, artifacts were removed by excluding events with extreme values of forward scatter (FSC.A and FSC.H) or fluorescence intensity (FL1.H and FL2.H). FL1.H corresponds to the height of the fluorescence signal acquired through the 510/10 filter, which is more sensitive to GFP emission, and FL2.H corresponds to the height of the fluorescence signal acquired through the 585/40 filter, which is more sensitive to YFP emission. Next, a principal component analysis was performed on the logarithm of FL1.H and FL2.H. The first principal component captured differences in fluorescence caused by variation in cell size, while the second component captured differences in fluorescence between cells expressing YFP and GFP. We then computed the Kernel density estimate of the second component, which allowed the separation of three populations of cells: (1) GFP cells with high scores on the second component, (2) YFP cells with low scores and (3) a smaller population with intermediate scores considered as doublets, *i.e.* events corresponding to two cells scored simultaneously, one expressing YFP and the other GFP. Doublets for which the two cells expressed the same fluorophore should also occur at low frequency in the GFP and YFP populations, but these doublets cannot be distinguished from singletons based on fluorescence. The number of YFP cells $N_Y$ and the number of GFP cells $N_G$ can be calculated from the total number of YFP events $T_Y$, the total number of GFP events $T_G$, the number of YFP doublets $D_Y$, the number of GFP doublets $D_G$ and the number of YFP-GFP doublets $D_{YG}$ using the following equations:

$$N_Y = T_Y + D_Y + D_{YG} \tag{1}$$

$$N_G = T_G + D_G + D_{YG} \tag{2}$$

By analogy with the Hardy-Weinberg principle, we could expect that:

$$D_{YG} = 2 \times \sqrt{D_Y} \times \sqrt{D_G} \tag{3}$$

Therefore,

$$D_Y = \frac{D_{YG}}{2 \times \sqrt{\frac{D_G}{D_Y}}} \tag{4}$$

If we assume that doublets were formed randomly, then we should expect the same proportion of doublets in the YFP and GFP populations:

$$\frac{D_Y}{T_Y} = \frac{D_G}{T_G} \tag{5}$$

We can deduce from
*Equations (1), (4) and (5)* that:

$$N_Y = T_Y + \frac{D_{YG}}{2 \times \sqrt{\frac{T_G}{T_Y}}} + D_{YG} \qquad (6)$$

Similarly,

$$N_G = T_G + \frac{D_{YG}}{2 \times \sqrt{\frac{T_Y}{T_G}}} + D_{YG} \qquad (7)$$

As all variables in the right-hand sides were known, we used *Equations (6) and (7)* to estimate the number of YFP cells and the number of GFP cells in the sample. Then, for each sample, we determined the number of cell generations that occurred during the three dilution cycles, by using the measured cell densities before each dilution as well as the dilution factor. The median number of generations for all samples grown on a same 96-well plate was used as a rough estimate of the number of generations for the samples of the plate. The number of generations over the entire experiment was found to be about 22. The fitness of the YFP strain relative to the GFP competitor was calculated as the exponential of the slope of $\log_e(N_Y/N_G)$ regressed on the number of generations at the four time points when genotype frequency was measured, based on equation (5.3) in *Cormack et al. (1990)*. For each tested strain, samples for which fitness departed from the median fitness among all eight replicate populations by more than five times the median absolute deviation were considered outliers and were excluded from further analysis. Outliers could occur for several reasons, one of them being the random appearance of a beneficial (or compensatory) de novo mutation during competitive growth (*Gallet et al., 2012*). For each sample, the fitness relative to the GFP strain was then divided by the mean fitness obtained for all replicate populations of the reference strain YPW1189 (for single-copy $P_{TDH3}$ variants) or YPW2682 (for double-copy $P_{TDH3}$ variants). We then calculated the mean relative fitness and standard deviation over the eight replicate populations of each tested strain. This measure of fitness expressed relative to a strain with the reference *TDH3* promoter sequence was used in all subsequent analyses. Processed fitness data are available in *Supplementary file 1* – Dataset 3 and *Source data 1*.

## Pairwise competition assays

To directly determine how expression noise impacts fitness at different levels of *TDH3* expression, we competed five pairs of strains with similar average expression levels but differences in *TDH3* expression noise, with each pair having a different average expression level (*Figure 4—source data 1*). This experimental design allowed differences in fitness caused by variation in noise to be directly observed without the assumption of transitivity (*Gallet et al., 2012*) and without the need to correct for the correlation between median expression and noise. In these experiments, we doubled the number of replicate populations (16) and the number of generations of growth (~42) to achieve greater precision in the fitness estimates for each pair of strains. We also measured the relative frequency of the two competitors using quantitative pyrosequencing instead of flow cytometry. This method did not require the expression of different fluorescent markers to distinguish cells from the two strains, allowing us to compete strains that only differed genetically by the mutations in the *TDH3* promoter affecting expression noise.

The competition assays were performed as described above, with the following differences in the protocol. First, strains with low noise and strains with high noise were arrayed each on two 96-well plates, with 16 replicates per genotype. After incubation on YPG omnitrays and growth to saturation in YPD, equal volumes of cultures of strains with low noise and high noise were mixed together and diluted in 0.5 ml of YPD. Following 36 hours of acclimation (as described above), six cycles of dilution followed by 12 hours of growth were performed. Dilution factors were calculated so that the average cell density on each plate would reach ~5 x $10^6$ cells/ml at the next dilution time point. Once every two cycles, the remaining cell cultures were centrifuged immediately after dilution and the pellets were stored in 30 µl of water at -80°C for later PCR amplification and pyrosequencing, so that genotype frequencies were quantified at four time points during the experiment. Cell pellets were thawed in the week after freezing and 15 µl of each sample was transferred in 30 µl of Zymolyase 20T (3 mg/ml) in 0.1M Sorbitol. After 15 minutes of incubation at 37°C, plates were vortexed vigorously for 15 seconds to disrupt cell wall and centrifuged at 3220 rcf for four minutes in an

Eppendorf 5810 R centrifuge. 5 µl of supernatant were used as PCR template in 50 µl reactions that also included 1 µl of dNTPs (10 mM of each dNTP), 2.5 µl of forward and reverse primers (10 µM each), 10 µl of 5x KAPA2G Buffer B, 0.4 µl of KAPA2G Robust HotStart DNA polymerase (5U/µl) and 28.6 µl of water. The oligonucleotides used for each sample were specific to the target mutation in $P_{TDH3}$ (*Supplementary file 2*). After 42 cycles of amplification, PCR products were denatured and purified using a PyroMark Q96 Vacuum Workstation (Qiagen) and pyrosequencing was performed on a PyroMark Q96 ID instrument using sequencing primers specific to the target mutations (*Figure 4—source data 1* and *Supplementary file 2*). The frequency of the two genotypes was determined for each sample from the relative heights of the peaks corresponding to the two alternative nucleotides on the pyrograms. Samples with an average peak height below 5 were excluded, as this could result from weak PCR and cause biases in measured allele frequency. The number of generations between each time point was estimated using the cell densities before each dilution and the dilution factors as described above. Relative fitness was calculated as the exponential of the slope of $\log_e(f_H/f_L)$ regressed on the number of generations across the four time points, where $f_H$ and $f_L$ were the relative frequency of the genotypes conferring high and low noise respectively ($f_H + f_L = 1$). Therefore, a fitness value above 1 meant that the strain with high noise grew faster than the strains with low noise, while a fitness value below 1 meant that the strain with low noise grew faster than the strain with high noise. For each pair of strains, replicates for which fitness departed from the median fitness across all 16 replicates by more than five times the median absolute deviation were considered outliers and were excluded. Fitness data for these direct competition assays are available in *Supplementary file 1* – Dataset 5 and *Figure 4—source data 1*.

## Analyzing the relationship between expression and fitness

The relationship between the average expression level of *TDH3* and fitness is not expected to follow a simple mathematical function. Therefore, we used LOESS regression to describe the relationship between median expression and fitness from the data collected with the set of 43 $P_{TDH3}$ alleles, using the R function *loess* with a span of 2/3. Next, we tested the impact of expression noise on fitness, which was complicated by the fact that expression noise is correlated with median expression and by the fact that median expression is expected to have a larger impact on fitness than expression noise. To disentangle the effects of median expression and noise on fitness, we first calculated the residuals (ΔNoise) from a LOESS regression (span = 2/3) of expression noise on median expression. Next, we used a similar approach to calculate the residuals (ΔFitness) from a LOESS regression (span = 2/3) of fitness on median expression. ΔFitness is the variation in fitness that cannot be explained by a difference in median expression in our dataset. To test whether ΔFitness could be at least partially explained by variation in expression noise, we calculated the Pearson's correlation coefficient between ΔNoise and ΔFitness and used the R function *cor.test* to test for significance of this correlation. We excluded the two strains that showed a median expression level above 125%, because the number of samples with high expression was too low for meaningful interpretation of ΔNoise and ΔFitness in this range of expression levels. In addition, we compared the correlations between ΔNoise and ΔFitness for two different classes of promoter variants determined based on their expression levels. First, we determined the maximum fitness from the LOESS regression of fitness on median expression. Next, we estimated the median expression value that would lead to a 0.005 reduction in fitness relative to the maximum. This expression value was used as a threshold to determine which strains had an expression close to the optimum or far from it. Three quantitative parameters were determined arbitrarily in these analyses: the span of the two LOESS regressions and the reduction in fitness used to determine the expression threshold. To test the robustness of the results to variation in these parameters, we calculated the correlations between ΔNoise and ΔFitness for 100 combinations of parameters where the span of the LOESS regressions took one of five values (2/6, 3/6, 4/6, 5/6 and 1) and the reduction in fitness took one of four values (0.0025, 0.005, 0.0075 and 0.01). In addition, we used permutation tests to compare median expression, ΔNoise values and ΔFitness values between two groups of genotypes: those with ΔNoise values below −1 (low noise) and those with ΔNoise values above 1 (high noise). For each parameter considered (median expression, ΔNoise and ΔFitness), the observed values were randomly shuffled between the two groups 100,000 times. P-values were then calculated as the proportion of shuffled groups for which the absolute difference of median was greater than the observed difference of

median between the groups before shuffling. All analyses were repeated for the four different metrics of noise mentioned above (Noise strength, SD, CV* and log(CV*)).

## Expression and fitness measured using TDH3-YFP fusion proteins

One important assumption in our analyses of the relationship between *TDH3* expression and fitness is that the median and noise of expression measured using the fluorescent reporter constructs inserted at *HO* are representative of the expression level of the TDH3 protein when the promoter variants are introduced at the native *TDH3* locus. To test whether the effects of mutations in the *TDH3* promoter were the same when introduced at *HO* or at the native *TDH3* locus, we constructed a *TDH3-YFP* fusion gene at the *TDH3* locus and then introduced 20 different $P_{TDH3}$ alleles upstream of this reporter gene, including eight TFBS and four TATA box variants that were present in the competition assays (*Figure 2—figure supplements 1—source data 1*). To fuse the coding sequences of *TDH3* and *YFP*, we amplified the *YFP-T$_{CYC1}$-KanMX4* construct from strain YPW1002 using primers 3415 and 3416 and transformed the PCR product in the non-fluorescent strain YPW978. Primer 3415 was designed to remove the stop codon of *TDH3* and the start codon of *YFP* and to insert a 30 bp linker between the coding sequences of the two genes (*Huh et al., 2003*). Then, the *TDH3* promoter was replaced with a *CORE-UH* cassette (*URA3-HphMX4*) amplified with oligonucleotides 1909 and 1910 (*Supplementary file 2*) to create strain YPW1618. The 20 $P_{TDH3}$ alleles were amplified from the native locus in the strains previously constructed (*Figure 2—figure supplements 1—source data 1*) using oligonucleotides 1344 and 1342 (*Supplementary file 2*) and transformed into YPW1618 to replace the *CORE-UH* cassette. The presence of the expected mutations was confirmed by sequencing PCR products obtained with primers 1345 and 1952 (*Supplementary file 2*). The fluorescence level of the strains expressing the fusion proteins was measured in parallel to the fluorescence of strains carrying the same $P_{TDH3}$ alleles at the *HO* locus. Four replicate samples of each strain were analyzed by flow cytometry after growth in YPD medium as described above. The expression of the reporter gene at the *HO* locus was found to be a strong predictor of the expression of the gene fusion at the native *TDH3* locus, both for median expression level (*Figure 2—figure supplement 1A*, $R^2 = 0.99$) and for expression noise (*Figure 2—figure supplement 1B*, $R^2 = 0.76$). These expression data are available in *Supplementary file 1 – Dataset 4* and *Figure 2—figure supplements 1—source data 1*. Flow data (FCS files) used to compare the effects of the $P_{TDH3}$ alleles when inserted at the *HO* locus and at the native *TDH3* locus are available in the FlowRepository (flowrepository.org) under experiment ID FR-FCM-ZYJX. In addition, the impact of fusing YFP to TDH3 on fitness was quantified by comparing the competitive growth rate of strain YPW1002 expressing YFP from the *HO* locus to the growth rate of strain YPW1964 expressing the TDH3-YFP protein fusion. The expression of the fusion protein was found to cause a 2.5% reduction in fitness (*Figure 2—figure supplement 1C*), which could be caused by altered function and/or stability of the TDH3 protein when fused with YFP. For this reason, we decided not to use protein fusions to measure the fitness associated with different levels of TDH3 expression.

## Modeling the relationship between single cell expression level and population fitness

To understand how cell-to-cell variability in gene expression level could contribute to population fitness, we performed individual-based stochastic simulations of the growth of clonal populations of cells covering a wide range of mean expression and expression noise values of a single gene. All simulations were run as short experiments of fixed duration (1,000 minutes) where variability in expression level impacting single cell division rate was the only determinant of population growth rate. The behavior of the population was determined by: a) a normal distribution $N_E$ of expression levels for the focal genotype described by its mean $\mu_E$ and variance $\sigma_E^2$, and b) a function $DT = f(E)$ relating single cell expression level $E$ to the time in minutes separating two consecutive cell divisions, or doubling time $DT$. Single cell expression levels sampled from the expression distribution defined the doubling time for a given cell. Two different functions relating expression level to $DT$ were explored: 1) a linear function ($DT = -40 \times E + 160$) and 2) an inverted Gaussian function ($DT = -160 \times \exp\left(-(E-1)^2/0.18\right) + 240$). In each run of the simulation, a population of cells was tracked by recording information on the current expression level of each cell, the current $DT$ derived from that expression level, and the amount of time remaining before the end of the

experiment. For simplicity, the expression level of each mother and daughter cell was drawn from the normal distribution $N_E$3 at each cell division and this expression level directly determined the $DT$3 value for the cell. To seed a starting population, $10^3$ cells were sampled from the expression distribution and their expression level was transformed into $DT$3. To desynchronize the founding population, the initial values of $DT$3 were scaled by a random value between 0 and 1 to randomize the time to first division and a complete simulation was run. $10^3$ cells were drawn randomly at the end of the seed experiment and used to found a population for which growth rate was quantified. In the body of the simulation, each single cell was evaluated to determine if the current $DT$3 was greater than the remaining time in the experiment assessed for that cell, and if so, the cell divided, at which point new expression levels were drawn randomly from the normal distribution $N_E$3 and independently for the mother and daughter cells. After cell division, the time remaining in the experiment for both mother and daughter cells decreased by the amount of the last $DT$3, the new expression levels were translated into new values of $DT$3, and the process repeated until $DT$3 values for all cells were greater than the remaining time in the experiment. Competitive fitness was calculated from the ratio of total number of cells $N_i$3 at the end of the experiment and the total number of cells $N_{ref}$3 obtained from simulating the growth of a reference genotype with mean expression $\mu_{ref} = 13$ and noise $\nu_{ref} = 0.13$, as follows:

$$Fitness = \exp\left(\frac{\ln\left(\frac{N_i}{N_{ref}}\right)}{T}\right)$$

Mean expression $\mu_E$ of experimental genotypes were explored in the interval [0,2]. Noise values $\nu_E$ were explored in the interval [0, 3] where noise was specified separately as standard deviation, coefficient of variation, and Fano factor. Experiment duration $T$ was set at 1000 minutes for ease of computation. 100 replicates of each stochastic simulation were run to estimate 95% confidence intervals on fitness estimates. Simulations were coded in MATLAB R2015 (*Supplementary file 5*).

## Measuring single-cell division rates

We performed time-lapse imaging of cells grown in microfluidic devices to compare the distributions of doubling time among four strains chosen for their differences in *TDH3* mean expression level and expression noise (strains YPW2879, YPW2868, YPW3064 and YPW3047 in *Figure 4—source data 1*). The four strains were assayed on four consecutive days using the same procedure. First, cells were grown to saturation in liquid YPD medium at 30°C for ~16 hr. Then, 100 μl of culture was transferred in 5 ml of fresh YPD and grown for another 4 hr at 30°C until it reached an optical density at 660 nm comprised between 0.2 and 0.3 (~3×10⁶ cells/ml). At this point, ~100 μl of cell culture was injected in a microfluidic chip using a 1 ml syringe. Microfluidic devices consisted of a PDMS (Polydymethylsiloxane) chip mounted on a 24 mm x 60 mm coverslip, as described in *Llamosi et al., 2016*. Each device contained five imaging chambers of 200 × 200×3.7 μm where a monolayer of cells could be grown. These chambers were connected on two sides to wide flow channels of 100 μm height where YPD medium was allowed to flow at 120 ul/min using an Ismatec IPC tubing pump. Cells were imaged using an inverted microscope (Olympus IX83) equipped with a CoolLED pE-300 illumination system, a Zyla sCMOS camera (Andor) and an IX3-ZDC2 system for autofocus. The temperature of the entire microfluidic system was maintained at 30°C in a Plexiglass chamber covering the microscope (Life Imaging Services). After 60 min of acclimation to growth in the microfluidic device, one bright field image and one fluorescence image were recorded once every six minutes for twenty hours at five positions centered on each of the five imaging chambers using a 60x oil immersion objective (Olympus PlanApo N 60x). Only images obtained during the first eight hours (80 frames) were analyzed, because tracking was not reliable after this time because of high cell densities. Fluorescence images captured expression from the $P_{TDH3}$-*YFP* reporter gene in each strain with a wild-type TDH3 promoter that was used for cell counting in the fitness assays. We were unable to reliably track individual cells and to correctly assign buds to their mother cells with a software for automated image analysis (ilastik v1.3.0) using this cytoplasmic YFP expression, thus we measured doubling times by analyzing the bright field images manually with Fiji (*Schindelin et al., 2012*). Raw bright field and fluorescence images, as well as bright field images where cell division events were annotated, are available on Zenodo (https://zenodo.org) with DOI 10.5281/zenodo.1327545. For each

movie, we randomly selected eight cells on the first frame and determined the doubling times of all cells produced by these eight starting cells. The doubling time of a cell was defined as the time separating the appearance of two consecutive buds (*Figure 5—figure supplement 1*). Following this procedure, we quantified the doubling time of at least 362 cells for each of the four genotypes. We then compared the mean doubling time and the standard deviation of doubling time for pairs of genotypes using permutation tests in R: doubling time values of the two genotypes were pooled and resampled $10^5$ times without replacement in two groups of same size as the number of cells analyzed for the two genotypes. P-values were calculated as the proportion of permutations for which the absolute difference of mean doubling time (or standard deviation) between the two groups was greater than the observed absolute difference between the two genotypes.

## Computational analyses

Custom R scripts containing the code used to process and analyze data as described above are provided as *Supplementary file 3*. Input files necessary to run the R scripts are available as. zip files in *Supplementary file 4*. Matlab code used to model population growth is provided as *Supplementary file 5*.

## Acknowledgements

We thank Pascal Hersen for the use of a time-lapse microscope, Lisa Kim for help determining single-cell division rates, and all members of the Wittkopp lab for helpful comments on the manuscript. This work was supported by a European Molecular Biology Organization postdoctoral fellowship (EMBO ALTF 1114–2012) to FD, University of Michigan Rackham Graduate School (BPHM), National Institutes of Health Genome Sciences training grant (T32 HG000040 to BPHM), National Institutes of Health National Research Service Award (1F32GM115198) to AH-D, and grants from the National Science Foundation (MCB-1021398) and National Institutes of Health (R01GM108826 and 1R35GM118073) to PJW.

## Additional information

### Competing interests

Patricia J Wittkopp: Senior editor, *eLife*. The other authors declare that no competing interests exist.

### Funding

| Funder | Grant reference number | Author |
| --- | --- | --- |
| European Molecular Biology Organization | EMBO ALTF 1114-2012 | Fabien Duveau |
| National Institutes of Health | 1F32GM115198 | Andrea Hodgins-Davis |
| National Institutes of Health | T32 HG000040 | Brian PH Metzger |
| National Institutes of Health | R01GM108826 | Patricia J Wittkopp |
| National Science Foundation | MCB-1021398 | Patricia J Wittkopp |
| National Institutes of Health | R35GM118073 | Patricia J Wittkopp |

The funders had no role in study design, data collection and interpretation, or the decision to submit the work for publication.

### Author contributions

Fabien Duveau, Conceptualization, Data curation, Software, Formal analysis, Supervision, Funding acquisition, Validation, Investigation, Visualization, Methodology, Writing—original draft, Writing—review and editing; Andrea Hodgins-Davis, Conceptualization, Resources, Data curation, Software, Formal analysis, Validation, Investigation, Visualization, Methodology, Writing—original draft, Writing—review and editing; Brian PH Metzger, Conceptualization, Resources, Methodology, Writing—review and editing; Bing Yang, Resources, Investigation, Writing—review and editing; Stephen

Tryban, Resources, Investigation; Elizabeth A Walker, Investigation, Project administration; Tricia Lybrook, Resources, Investigation, Contributed by constructing the strains included in Figure 2 - figure supplement 1, Read and approved the submitted version of the manuscript; Patricia J Wittkopp, Conceptualization, Supervision, Funding acquisition, Visualization, Methodology, Writing—original draft, Project administration, Writing—review and editing

### Author ORCIDs
Fabien Duveau (iD) http://orcid.org/0000-0003-4784-0640
Patricia J Wittkopp (iD) http://orcid.org/0000-0001-7619-0048

### Decision letter and Author response
Decision letter https://doi.org/10.7554/eLife.37272.047
Author response https://doi.org/10.7554/eLife.37272.048

# Additional files

### Supplementary files
• Supplementary file 1. Datasets generated using R scripts available in *Supplementary file 3* and used to make Source Data files for figures.
DOI: https://doi.org/10.7554/eLife.37272.031

• Supplementary file 2. List of oligonucleotides used in this study.
DOI: https://doi.org/10.7554/eLife.37272.032

• Supplementary file 3. R scripts for the analysis of pyrosequencing and flow cytometry data used to determine the median expression, expression noise and fitness associated with different $P_{TDH3}$ alleles.
DOI: https://doi.org/10.7554/eLife.37272.033

• Supplementary file 4. Zip folder containing all input files necessary to run the R scripts available in *Supplementary file 3*.
DOI: https://doi.org/10.7554/eLife.37272.034

• Supplementary file 5. Matlab code used to model the growth of cell populations with different levels of mean expression and expression noise.
DOI: https://doi.org/10.7554/eLife.37272.035

• Source data 1. Expression and fitness data for the final set of 43 *TDH3* promoter alleles. Fluorescence levels and competitive fitness were measured by flow cytometry and analyzed using the R script provided in *Supplementary file 3*. Data used to make *Figure 1D*, *Figure 2C* and *Figure 3*.
DOI: https://doi.org/10.7554/eLife.37272.036

• Transparent reporting form
DOI: https://doi.org/10.7554/eLife.37272.037

### Data availability
Flow data (FCS files) used to quantify fluorescence levels produced by the 43 TDH3 promoter alleles are available in the FlowRepository (flowrepository.org) under experiment ID FR-FCM-ZY8Y. Raw bright field and fluorescence images, as well as bright field images where cell division events were annotated, are available on Zenodo (https://zenodo.org) with DOI 10.5281/zenodo.1327545. All other data are provided as source data and/or supplementary files with the manuscript.

The following datasets were generated:

| Author(s) | Year | Dataset title | Dataset URL | Database, license, and accessibility information |
| --- | --- | --- | --- | --- |
| Fabien Duveau | 2018 | Time-lapse images of yeast cells with different levels of TDH3 median expression and expression noise | https://dx.doi.org/10.5281/zenodo.1327545 | Publicly available at Zenodo (https://zenodo.org/). |
| Fabien Duveau, Patricia J Wittkopp | 2016 | Activity of 48 TDH3 promoter variants in YPD | http://flowrepository.org/id/FR-FCM-ZY8Y | Publicly available at FlowRepository |

| | | | | |
|---|---|---|---|---|
| | | | | (accession no. FR-FCM-ZY8Y) |
| Fabien Duveau, Patricia J Wittkopp | 2016 | Effects of 20 TDH3 promoter alleles at HO and native TDH3 loci | http://flowrepository.org/id/FR-FCM-ZYJX | Publicly available at FlowRepository (accession no. FR-FCM-ZYJX) |
| Fabien Duveau, Patricia J Wittkopp | 2016 | Fitness of 48 TDH3 promoter alleles in YPD | http://flowrepository.org/id/FR-FCM-ZYJN | Publicly available at FlowRepository (accession no. FR-FCM-ZYJN) |

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
