## [Decision Letter]

Thank you for submitting your article "Fitness effects of altering gene expression noise in *Saccharomyces cerevisiae*" for consideration by *eLife*. Your article has been reviewed by Naama Barkai as the Senior Editor, a Reviewing Editor, and two reviewers. The following individual involved in review of your submission has agreed to reveal his identity: Kevin J Verstrepen (Reviewer #2).

The reviewers have discussed the reviews with one another and the Reviewing Editor has drafted this decision to help you prepare a revised submission.

Summary:

As you will see below, both reviewers liked the work. They appreciated the concept and recommended the experiments. Regarding the conclusions, the reviewers felt that they were rather expected, somewhat reducing enthusiasm.

The most important shared concern is whether the mutants used in this study offer a fair comparison to natural variants with different noise levels. The potential discrepancy between artificial and natural variants seems difficult to avoid, but at the very least, it should be acknowledged and critically discussed. We do recommend that you will perform the experiments suggested which relate to this point (in particular, measuring single-cell growth). Both reviewers felt that they will significantly upgrade the work, although they did not consider it as a must for publication. Critical discussion of this point was considered to be essential.

*Reviewer #1*:

The manuscript by Duveau et al. examines the impact of expression levels and noise in synthetic mutants of *TDH3*. This group has previously demonstrated that noise in *TDH3* expression is under purifying selection in wild yeast (Metzger et al., 2015), and this paper follows up on this finding by showing that, at least for synthetic mutants in a stable environment, the fitness effect of expression noise depends on the mean expression level and vice versa. Higher noise is generally beneficial when *TDH3* expression level is far from the optimum, but deleterious when near the optimum. The study is technically rigorous and carefully performed, and the results, although expected from previous theoretical and experimental work in expression noise and bet hedging, do experimentally demonstrate the important interdependence between mean expression level and noise on fitness in a stable environment.

Essential revisions:

1) The authors select 43 of 171 constructs generated to study further based on the expression level and expression noise profiles. However, the reader never sees the data for the full set and it is difficult to determine if the chosen set is representative. This should be included in a supplemental figure.

2) Naturally occurring variants of in the *TDH3* promoter were not studied here because the authors report that they strongly co-vary, making it difficult to separate the fitness effects of mean expression level and noise. Shouldn't the relationship of covariance of these variants, which have seen natural selection, be different from the designed mutations used here? Based on the results presented here, one would predict that the LOESS curves of the relationship between median expression and noise would have different patterns, with few naturally occurring variants displaying both low expression and low noise. Along similar lines, it appears from previous work (Metzger et al., 2015) that naturally occurring variation appears to only increase expression noise. Is this only happening when *TDH3* expression is far from the optimum?

3) More generally, I am having trouble connecting the results presented here to naturally occurring expression noise. The authors have convincingly demonstrated that higher noise is favored in a stable environment when the mean expression is far from the optimum. Based on previous theoretical and experimental work, this was expected. However, does this have anything to do with why expression noise varies across genotypes or genes? At least two other possibilities exist: (1) the fitness effects of changes in noise are not strong enough to be subject to selection (Metzger et al., 2015, shows this is likely not to be true for *TDH3*), or (2) higher noise is more fit in fluctuating environments. A reasonable hypothesis is that optimal expression levels of *TDH3* (involved in glycolysis and gluconeogenesis) would be highly dependent on nutrient availability of the environment, and that nutrient availability could change quickly, making high noise in expression adaptive. This hypothesis is supported by a number of studies that find bet-hedging-like mechanisms are playing out during nutrient shifts. Yet, this possibility is not directly addressed in this manuscript, rather the entire manuscript assumes that a stable environment with respect to *TDH3* expression can be reached. Thus, beyond careful experimental validation of an expected result, I am unsure what I have learned from this work. Comparisons between synthetic mutants and naturally occurring variation or across environmental fluctuations may be needed to increase impact.

*Reviewer #2*:

This study measures the fitness effect of a set of 236 mutant alleles of the *TDH3* promoter in *S. cerevisiae*. YFP fusions were used to estimate *TDH3* transcriptional activity and expression noise. Some of the mutants showed a different average expression of *TDH3*, some showed a different noise level, and most showed a combination of these effects. Comparing the fitness effects and transcription phenotype of different mutants indicates that high expression noise is detrimental when the average expression is close to the optimum/wild-type level, whereas increased noise has a positive fitness effect when the average expression levels deviate from the optimal/wild-type levels. Together, these results further confirm earlier studies that suggest that noise can help to diminish the negative fitness effects associated with a suboptimal (average) expression level because a fraction of the population will have a more optimal expression level (and this faster-growing fraction drives the average growth rate up), while on the other hand, noise may be selected against in populations where average expression is near-optimal.

Despite a few issues that merit a response from the authors (indicated below), I really like this study because its elegant and simple design provides leads to a clear conclusion that helps to answers an important question. Specifically, I appreciate the efforts to provide solid experimental data and the fair discussion of the results. Moreover, the text is particularly well-written.

I see four issues that perhaps should be addressed or at least acknowledged in the Discussion section.

Firstly, one can never be completely sure that the promoter variants that are generated offer a good and fair comparison to naturally-occurring variation in expression noise. For example, in theory, it is possible that some promoter/TATA box mutations might have pleiotropic fitness effects. I understand that this is difficult to avoid, and that the use of many different mutants at least partly mitigates this worry, but perhaps it should be acknowledged in the Discussion section.

Second, whereas widely accepted as a good method to measure expression noise, using promoter-YFP fusions as a proxy for expression/mRNA abundance inevitably has some shortcomings. I don't think it necessary to directly measure RNA levels in single cells across all the mutants, but perhaps a few mutants could be assayed to confirm that expression is accurately estimated? Moreover, the YFP fusions may also not perfectly align with Tdh3 protein levels (for example if Tdh3 levels would also be controlled post-transcriptionally). Making Tdh3-YFP fusions would only partly resolve this issue, as the YFP tag may influence Tdh3 stability. While these effects are again difficult to avoid and are unlikely to cause serious artefacts, it would be good to mention the issue in the discussion.

Third, it is a pity that the authors do not directly link single-cell expression levels with fitness. Instead, they use a model that tried to link a cell's *TDH3* expression to its replicative fitness. Using Tdh3-YFP protein fusions combined with single-cell monitoring technologies (e.g. simply using time-lapse microscopy to evaluate the growth rate a few single cells as they grow into microcolonies or using flow cells such as the commercially available CellAsic system), it should be relatively easy to directly measure the replication rate of cells and correlate this with the individual *TDH3* expression level (as estimated by YFP fluorescence). This would really tie the study together and allow linking noise to individual fitness to population-level fitness (as also suggested by the authors in the Conclusions section).

Fourth, is the number of cells with average (population-level) expression close to the optimum large enough to draw reliable conclusions? Given the considerable variation in the relation between fitness and mean expression level observed in the (much larger) set of mutants where average expression is not optimal, my gut feeling dictates some caution.

---

## [Author Response]

Summary:As you will see below, both reviewers liked the work. They appreciated the concept and recommended the experiments. Regarding the conclusions, the reviewers felt that they were rather expected, somewhat reducing enthusiasm.The most important shared concern is whether the mutants used in this study offer a fair comparison to natural variants with different noise levels. The potential discrepancy between artificial and natural variants seems difficult to avoid, but at the very least, it should be acknowledged and critically discussed. We do recommend that you will perform the experiments suggested which relate to this point (in particular, measuring single-cell growth). Both reviewers felt that they will significantly upgrade the work, although they did not consider it as a must for publication. Critical discussion of this point was considered to be essential.

We thank the reviewers for their helpful comments, which we address below in turn.

1) We agree that our findings are consistent with the benefits of noise in changing environments discussed in the bet hedging literature (and say so in the Conclusions section), but we think there is novelty in describing beneficial effects of noise in the absence of environmental fluctuations. To the best of our knowledge, such effects have not previously been observed or predicted outside of the theoretical papers we cite. (If you can point us to studies we missed, we’d be happy to cite them.) Perhaps more importantly, we think that even if variability in the fitness effects of noise caused by differences in mean expression were anticipated in a constant environment, empirically demonstrating these effects – and quantifying the fitness effects of noise more generally – still provides a significant advance for the field.

2) We agree that the mutant alleles used to disentangle fitness effects of mean and noise have different properties than natural variants: selection has limited variation in noise among naturally occurring alleles (Metzger et al., 2015) and mean expression in all natural variants tested is close to the expression optimum (Duveau et al., 2017). Although most of the mutants analyzed in the current study differ from naturally occurring alleles by only a few base-pairs or one duplication event, we do not expect most of these alleles to survive in natural populations because of their effects on mean expression and/or expression noise. Using these mutants was necessary, however, to quantify the effects of noise on fitness.

To make these differences between natural variants and mutant alleles clearer to readers, we have added the following sentences to subsection “Generating cariation in expression noise independent of average expression level” describing the construction of alleles in the revised version:

“These mutant alleles captured a much greater range of mean expression and expression noise than *TDH3* promoter alleles segregating in natural populations (Metzger et al., 2015; Duveau et al., 2017a) and allowed us to more fully explore the relationship between mean, noise and fitness than would be possible using naturally occurring variation alone.”

We have also added the following text to the Conclusions section to give a more concrete example of how we see our observations potentially affecting evolution of gene expression in natural populations:

“Our data suggest that high levels of expression noise can also be beneficial in a stable environment when the mean expression level is far from optimal. […] The relative frequency by which evolution proceeds through these two paths will depend on both the relative frequency of alleles that increase mean expression and expression noise, as well as the fitness differences between these alleles.”

3) We agree with the reviewers that directly relating single-cell expression and single-cell growth rates is a critical next step for the field. In fact, the first author has spent most of the last year in another lab developing microfluidic and microscopy tools to perform this experiment properly. As described below in response to reviewer 2, we were unable to perform this experiment exactly as suggested; however, we were able to trace single-cell lineages using time-lapse microscopy and compare the distributions of single-cell division times between genotypes with different levels of *TDH3* expression noise. We observed a significantly greater standard deviation for single-cell division times in genotypes with higher expression noise. We were also able to detect the expected faster cell division rate for genotypes with mean expression closer to the optimum. These data are consistent with assumptions used in the computational model and are presented as a supplemental figure referenced in subsection “Simulating population growth reveals fitness effects of noise” as follows:

“Empirical measures of single-cell division rates were consistent with these elements of the model, showing more variable cell division times in genotypes with greater *TDH3* expression noise and shorter cell division times in genotypes with mean *TDH3* expression closer to the fitness optimum (Figure 5—figure supplement 1).”

Reviewer #1:[…] 1) The authors select 43 of 171 constructs generated to study further based on the expression level and expression noise profiles. However, the reader never sees the data for the full set and it is difficult to determine if the chosen set is representative. This should be included in a supplemental figure.

The full set of 171 alleles is shown in Figure 1—figure supplement 1. This is conveyed to the reader in the following statement in subsection “Generating variation in expression noise independent of average expression level”:

“From this collection of 171 *TDH3* promoter alleles (Figure 1—figure supplement 1, Figure 1—source data 2), we selected 43 alleles (Figure 1-3—source data 1) to study the fitness effects of changes in average expression level and expression noise of the native *TDH3* gene.”

To make it easier for readers to compare the 43 selected promoter alleles to the full set of 171, we changed the symbols used to represent the selected alleles in this supplemental figure in the revised version.

2) Naturally occurring variants of in the TDH3 promoter were not studied here because the authors report that they strongly co-vary, making it difficult to separate the fitness effects of mean expression level and noise. Shouldn't the relationship of covariance of these variants, which have seen natural selection, be different from the designed mutations used here? Based on the results presented here, one would predict that the LOESS curves of the relationship between median expression and noise would have different patterns, with few naturally occurring variants displaying both low expression and low noise. Along similar lines, it appears from previous work (Metzger et al., 2015) that naturally occurring variation appears to only increase expression noise. Is this only happening when TDH3 expression is far from the optimum?

We struggled to follow this comment because we did not discuss the naturally occurring alleles studied in Metzger et al. (2015) in the initial submission. Perhaps the reviewer was thinking of the 236 point mutations in the *TDH3* promoter that tended to increase expression noise (Metzger et al., 2015)? We mentioned that the strong covariance between mean and noise for these genotypes made them insufficient for measuring the fitness effects of noise.

The naturally occurring alleles were not used in the current study because of their low levels of variation in expression noise and mean expression. Converting fluorescence based measures of expression reported for naturally occurring alleles in Metzger et al. (2015) and Duveau et al. (2017) to the estimated mRNA expression levels used in the current study, showed that the 27 naturally occurring haplotypes studied all had mean expression levels between 80% and 140% of the wild-type allele and would thus be classified as “close to the optimum”. Because of this limited variation in mean and noise, it is not possible to assess the fitness effects of expression noise or the relationship between median expression and noise resulting from naturally occurring alleles with mean expression far from the optimum. Given their range of effects on median expression and expression noise, no significant fitness differences are expected to exist among natural *TDH3* promoter alleles, consistent with the hypothesis that these variants are well adapted for growth in the environment assayed and that they are not useful to investigate how variation in median expression and noise affects fitness.

3) More generally, I am having trouble connecting the results presented here to naturally occurring expression noise. The authors have convincingly demonstrated that higher noise is favored in a stable environment when the mean expression is far from the optimum. Based on previous theoretical and experimental work, this was expected. However, does this have anything to do with why expression noise varies across genotypes or genes? At least two other possibilities exist: (1) the fitness effects of changes in noise are not strong enough to be subject to selection (Metzger et al., 2015, shows this is likely not to be true for TDH3), or (2) higher noise is more fit in fluctuating environments. A reasonable hypothesis is that optimal expression levels of TDH3 (involved in glycolysis and gluconeogenesis) would be highly dependent on nutrient availability of the environment, and that nutrient availability could change quickly, making high noise in expression adaptive. This hypothesis is supported by a number of studies that find bet-hedging-like mechanisms are playing out during nutrient shifts. Yet, this possibility is not directly addressed in this manuscript, rather the entire manuscript assumes that a stable environment with respect to TDH3 expression can be reached. Thus, beyond careful experimental validation of an expected result, I am unsure what I have learned from this work. Comparisons between synthetic mutants and naturally occurring variation or across environmental fluctuations may be needed to increase impact.

The goal for this study was not to explain observed levels of variation in expression noise for one or all genes in natural populations, but rather to understand how expression noise impacts fitness under different scenarios. By doing so, we identified scenarios under which higher levels of noise are beneficial even without the environmental shifts required for bet-hedging. We believe this result is important for several reasons. First, the beneficial effect of noise has been mostly studied in the context of bet-hedging when the optimal expression level changes over time due to environmental fluctuations. To our knowledge, few theoretical studies described how expression noise impacted fitness when the expression optimum remained unchanged (Zhang et al., 2009; Tǎnase-Nicola and ten Wolde, 2008) and our study is the first one providing experimental demonstration of the beneficial effect of noise far from the optimum. Second, we expect the beneficial (or detrimental) effect of increased noise to hold as long as the median expression level remains far from (or close to) optimum, which is a condition less restrictive and more realistic than a completely stable environment. For instance, the expression of several yeast genes was shown to remain close to optimal despite environmental change (Keren et al., 2016). A shift away from the optimal expression level may occur either through fixation of a deleterious allele by genetic drift or more likely by exposure to a novel environment that was rarely or never experienced in the past. Under such conditions, our results show that a change in median expression level is not the only beneficial option (as generally expected), but an increase in expression noise can also provide a selective advantage. Whether evolution proceeds through changes in median expression, expression noise or both depends not only on the fitness consequences of variation in median expression and expression noise, but also on the effects of new mutations on these two properties.

Determining to what extent differences in expression noise among genes is explained by selection in constant or fluctuating environments is difficult without knowing the past ecological history of natural populations. For *TDH3* specifically, as described above, comparing the effects of synthetic mutants and naturally occurring variation showed that selection has acted to minimize variation in expression noise (Metzger et al., 2015). This observation does not support the hypothesis proposed by the reviewer for which high noise in expression would be adaptive. However, our results are consistent with a scenario where median *TDH3* expression level is close to optimum for natural populations in their natural environments and therefore increased noise is counter-selected.

To try to make the implications of our findings for naturally occurring variation more concrete, we added this scenario to the Conclusions section:

“Our data suggest that high levels of expression noise can also be beneficial in a stable environment when the mean expression level is far from optimal. For example, if an allele driving suboptimally low expression were to be fixed in a population, selection should initially favor alleles that increase mean expression and/or expression noise. After alleles driving mean expression close to the optimum are fixed, selection should then favor alleles with lower levels of expression noise. The relative frequency by which evolution proceeds through these two paths will depend on both the relative frequency of alleles that increase mean expression and expression noise, as well as the fitness differences between these alleles.”

We also previously found evidence that the differences in median *TDH3* expression level observed after growth in different carbon sources was maintained by stabilizing selection, further suggesting that median expression was near the optimum despite changes in nutrient availability. This observation is now pointed out explicitly in the revised Conclusions section:

“Such plasticity in expression level seems to have already evolved for *TDH3* (Duveau et al., 2017).”

Reviewer #2:[…] I see four issues that perhaps should be addressed or at least acknowledged in the Discussion section.Firstly, one can never be completely sure that the promoter variants that are generated offer a good and fair comparison to naturally-occurring variation in expression noise. For example, in theory, it is possible that some promoter/TATA box mutations might have pleiotropic fitness effects. I understand that this is difficult to avoid, and that the use of many different mutants at least partly mitigates this worry, but perhaps it should be acknowledged in the discussion.

As described above, we have added a sentence to subsection “Generating variation in expression noise independent of average expression level” in the revised version explicitly stating that the mutants studied here have a different range of mean and noise levels than alleles segregating in natural populations. We have also acknowledged the potential effects of pleiotropy in the revised version by adding the following sentence to subsection “Fitness effects of changing average *TDH3* expression level”:

“We expect these differences in fitness among genotypes with different *TDH3* promoter alleles to arise primarily from differences in *TDH3* expression; however, differences in pleiotropy among promoter alleles might also contribute to differences in fitness.”

Second, whereas widely accepted as a good method to measure expression noise, using promoter-YFP fusions as a proxy for expression/mRNA abundance inevitably has some shortcomings. I don't think it necessary to directly measure RNA levels in single cells across all the mutants, but perhaps a few mutants could be assayed to confirm that expression is accurately estimated? Moreover, the YFP fusions may also not perfectly align with Tdh3 protein levels (for example if Tdh3 levels would also be controlled post-transcriptionally). Making Tdh3-YFP fusions would only partly resolve this issue, as the YFP tag may influence Tdh3 stability. While these effects are again difficult to avoid and are unlikely to cause serious artefacts, it would be good to mention the issue in the discussion.

We agree with the reviewer about limitations of using YFP fluorescence as a proxy for mRNA abundance and have taken steps to address these concerns. Although we did not perform single-cell RNA sequencing to compare to single-cell fluorescence, we did compare mean levels of YFP fluorescence to mean YFP mRNA abundance for populations of cells, as described in the methods section titled “Relationship between mRNA levels and fluorescence”. We found a non-linear relationship between fluorescence and mRNA abundance (shown in Figure 1C) and used this relationship to translate YFP fluorescence observed in each cell into estimated mRNA abundance for that cell. We believe that this is an important step toward capturing biological reality that we did not take in prior work.

We also used Tdh3-YFP fusion proteins to compare the effects of 20 *TDH3* promoter alleles at the native locus to the effects of these same alleles in the *P_TDH3_-YFP* reporter gene at the *HO* locus. These experiments are described in the “Expression and fitness measured using TDH3-YFP fusion proteins” section of the Materials and methods section. We found a near perfect correlation (R^2^=0.99) between fluorescence of the fusion and reporter genes, although the slope of this relationship was 0.83 rather than 1 (see Figure 2—figure supplement 1A). The strain carrying a Tdh3-YFP fusion protein under control of the wild-type *TDH3* promoter showed a 2.5% reduction in fitness (described in subsection “Expression and fitness measured using TDH3-YFP fusion proteins”), suggesting that the fusion might indeed have affected *TDH3* protein function or stability.

We have made the references in the main text to these key control experiments more explicit in the revised version of the manuscript by modifying text in subsection “Fitness effects of changing average TDH3 expression level” to read as follows:

“Expression of this reporter gene provided a reliable readout of average expression level and expression noise driven by the same *P_TDH3_*promoters at the native *TDH3* locus, as measured using Tdh3-YFP fusion proteins (Figure 2—figure supplement 1A, B). These fusion protein alleles were not used for comparing fitness effects among TDH3 promoter alleles because the YFP fusion reduced fitness by 2.5% relative to a strain expressing *TDH3* and YFP from independent promoters (Figure 2—figure supplement 1C).”

Third, it is a pity that the authors do not directly link single-cell expression levels with fitness. Instead, they use a model that tried to link a cell's TDH3 expression to its replicative fitness. Using Tdh3-YFP protein fusions combined with single-cell monitoring technologies (e.g. simply using time-lapse microscopy to evaluate the growth rate a few single cells as they grow into microcolonies or using flow cells such as the commercially available CellAsic system), it should be relatively easy to directly measure the replication rate of cells and correlate this with the individual TDH3 expression level (as estimated by YFP fluorescence). This would really tie the study together and allow linking noise to individual fitness to population-level fitness (as also suggested by the authors in the Conclusions section).

We agree that being able to directly link single-cell expression with single-cell division rates would be an excellent addition to this work, and we were able to collect new data that speaks to this point, as described below. We did not perform this experiment with Tdh3-YFP fusion proteins as suggested because (1) the 24 hour half-life of Venus YFP is expected to mask the types of fluctuations in expression that occur on the time-scales of cell division (1-3 hours) that we would want to measure, and (2) the fusion of YFP to Tdh3 caused a 2.5% reduction in fitness, suggesting that it interferes with *TDH3* function or stability. In addition, the set of 20 *TDH3* promoter alleles driving expression of a Tdh3-YFP fusion protein we previously constructed did not include pairs of alleles with similar mean expression but differences in expression noise. Our other strains were also not suitable for this experiment because they have mutations in the *TDH3* promoter upstream of only YFP (suitable for measuring effects on expression but not fitness) or only the native *TDH3* gene (suitable for measuring effects on fitness but not expression).

As an alternative to the proposed experiment, we used time-lapse microscopy to measure single-cell division rates in four genotypes grown in custom microfluidic devices. These four genotypes contained two pairs of genotypes with similar mean expression levels but different levels of expression noise. One of these pairs of genotypes had mean expression close to the optimum, whereas the other had mean expression far from the optimum. As shown in Figure 5—figure supplement 1, we found the genotype in each pair showing higher expression noise with the *P_TDH3_-YFP* reporter gene had significantly greater variation in single-cell division rates. We also observed significantly faster mean cell division rates for the pair of genotypes with mean expression closer to the fitness optimum, as expected. These data confirm key assumptions of our model and are described in subsection “Simulating population growth reveals fitness effects of noise” of the revised manuscript, as follows:

“Empirical measures of single-cell division rates were consistent with these elements of the model, showing more variable cell division times in genotypes with greater *TDH3* expression noise and shorter cell division times in genotypes with mean *TDH3* expression closer to the fitness optimum (Figure 5—figure supplement 1).”

Fourth, is the number of cells with average (population-level) expression close to the optimum large enough to draw reliable conclusions? Given the considerable variation in the relation between fitness and mean expression level observed in the (much larger) set of mutants where average expression is not optimal, my gut feeling dictates some caution.

We understand the reviewer’s concern, especially for the analysis shown in Figure 3D and E, which is most affected by the number of alleles in the “close to the optimum” and “far from the optimum” sets. We are encouraged, however, that additional support for our inference that selection disfavors higher noise when mean expression is near the optimum comes from the head-to-head comparisons shown in Figure 4 and Figure 5—figure supplement 1, the modeling results shown in Figure 6D, prior theoretical work by Tǎnase-Nicola and ten Wolde (2008), and empirical data showing evidence of selection reducing noise among naturally occurring alleles (Metzger et al., 2015) that appear to drive mean expression close to the optimum (Duveau et al., 2017a, 2017b). Nonetheless, we added the following sentence to subsection “Disentangling the effects of TDH3 expression level and expression noise on fitness” discussing results shown in Figure 3D and E in the revised version:

“We note, however, that the smaller number of genotypes with mean expression close to the optimum provided less power to detect a significant relationship than genotypes with mean expression far from the optimum. “